# Interval forecasts of weekly incident and cumulative COVID-19 mortality in the United States: A comparison of combining methods

**Kathryn S. Taylor** [1]*, **James W. Taylor** [2]

1 Nuffield Department of Primary Care Health Sciences, University of Oxford, Oxford, United Kingdom,
2 Saïd Business School, University of Oxford, Oxford, United Kingdom

* kathryn.taylor@phc.ox.ac.uk

## Abstract

### Background

A combined forecast from multiple models is typically more accurate than an individual forecast, but there are few examples of studies of combining in infectious disease forecasting. We investigated the accuracy of different ways of combining interval forecasts of weekly incident and cumulative coronavirus disease-2019 (COVID-19) mortality.

### Methods

We considered weekly interval forecasts, for 1- to 4-week prediction horizons, with out-of-sample periods of approximately 18 months ending on 8 January 2022, for multiple locations in the United States, using data from the COVID-19 Forecast Hub. Our comparison involved simple and more complex combining methods, including methods that involve trimming outliers or performance-based weights. Prediction accuracy was evaluated using interval scores, weighted interval scores, skill scores, ranks, and reliability diagrams.

### Results

The weighted inverse score and median combining methods performed best for forecasts of incident deaths. Overall, the leading inverse score method was 12% better than the mean benchmark method in forecasting the 95% interval and, considering all interval forecasts, the median was 7% better than the mean. Overall, the median was the most accurate method for forecasts of cumulative deaths. Compared to the mean, the median's accuracy was 65% better in forecasting the 95% interval, and 43% better considering all interval forecasts. For all combining methods except the median, combining forecasts from only compartmental models produced better forecasts than combining forecasts from all models.

### Conclusions

Combining forecasts can improve the contribution of probabilistic forecasting to health policy decision making during epidemics. The relative performance of combining methods depends on the extent of outliers and the type of models in the combination. The median

**Data Availability Statement:** Data were downloaded from the public GitHub data repository of the COVID-19 Hub at https://github.com/reichlab/covid19-forecast-hub. The code used to

generate the results is publically available on Zenodo at https://doi.org/10.5281/zenodo.6300524.

**Funding:** This research was partly supported by the National Institute for Health Research Applied Research Collaboration Oxford and Thames Valley at Oxford Health NHS Foundation Trust. The views expressed in this publication are those of the author(s) and not necessarily those of the NIHR or the Department of Health and Social Care. The funders had no role in study design, data collection and analysis, decision to publish, or preparation of the manuscript.

**Competing interests:** The authors have declared that no competing interests exist.

combination has the advantage of being robust to outlying forecasts. Our results support the Hub's use of the median and we recommend further investigation into the use of weighted methods.

## Introduction

The coronavirus disease-2019 (COVID-19) pandemic has overwhelmed health services and caused excess death rates, prompting governments to impose extreme restrictions in attempts to control the spread of the virus [1–3]. These interventions have resulted in multiple economic, health and societal problems [4, 5]. This has generated intense debate among experts about the best way forward [6]. Governments and their advisors have relied upon forecasts from models of the numbers of COVID-19 cases, hospitalisations and deaths to help decide what actions to take [7]. Using models to lead health policy has been controversial, but it is recognised that modelling is potentially valuable when used appropriately [1, 8–10]. Numerous models have been developed to forecast different COVID-19 data, e.g. [11–13].

Models should provide probabilistic forecasts, as point forecasts are inherently uncertain [9, 14]. A 95% interval forecast is a common and useful form of probabilistic forecast [15, 16]. Models may be constructed for prediction or scenario analysis. Prediction models forecast the most likely outcome in the current circumstances. Multiple models may reflect different approaches to answering the same question [11], and conflicting forecasts may arise. Rather than asking which is the best model [17], a forecast combination can be used, such as the mean, which is often used and hard to beat [18, 19]. Forecast combining harnesses the 'wisdom of the crowd' [20] by producing a collective forecast from multiple models that is typically more accurate than forecasts from individual models. Combining pragmatically synthesises information underlying different prediction methods, diversifying the risk inherent in relying on an individual model, and it can offset statistical bias, potentially cancelling out overestimation and underestimation [21]. These advantages are well-established in many applications outside health care [22–25]. This has encouraged the more recent applications of combining in infectious disease prediction [14, 26–29], including online platforms that present visualisations of combined probabilistic forecasts of COVID-19 data from the U.S, reported by the Centers for Disease Control and Prevention (CDC), and from Europe, reported by the European Centre for Disease and Control (EDCD). Other examples or combined probabilistic forecasts are in vaccine trial planning [30] and diagnosing disease [31]. These examples have mainly focused on simple mean and median 'ensembles' and, in the case of prediction of COVID-19 data, published studies have primarily involved short periods of data, which rules out the consideration of more sophisticated methods, such as those weighted by historical accuracy.

By comparing the accuracy of different combining methods over longer forecast evaluation periods compared to other studies, our broad aims were to: (a) investigate whether combining methods, involving weights determined by prior forecast accuracy or different ways of excluding outliers, are more accurate than simple methods of combining, and (b) establish the relative accuracy of the mean and median combining methods. Previously, we reported several new weighted methods, in a comparison of combining methods applied to probabilistic predictions of weekly cumulative COVID-19 mortality in U.S. locations over the 40-week period up to 23 January 2021 [32]. We found that weighted methods were the most accurate overall and the mean generally outperformed the median except in the first ten weeks. In this paper, we test further by comparing the combining methods on a dataset of interval forecasts of

cumulative mortality and a dataset of incident mortality, both over a period of more than 80 weeks. We also include individual models in the comparison, and explore the impacts of reporting patterns of death counts and outlying forecasts on forecast accuracy.

## Materials and methods

### Data sources

Forecasts of weekly incident and cumulative COVID-19 mortalities were downloaded from the COVID-19 Forecast Hub (https://covid19forecasthub.org/), which is an ongoing collaboration with the U.S. CDC, and involves forecasts submitted by teams from academia, industry and government-affiliated groups [26]. Teams are invited to submit forecasts for 1- to 4-week horizons, in the form of a point forecast and estimates of quantiles corresponding to the following 23 probability points along the probability distribution: 1%, 2.5%, 5%, 10%, 15%, 20%, 25%, 30%, 35%, 40%, 45%, 50%, 55%, 60%, 65%, 70%, 75%, 80%, 85%, 90%, 95%, 97.5% and 99%. From these, we produced interval forecasts, including the 95% interval, which is bounded by the 2.5% and 97.5% quantiles. The numbers of actual cumulative COVID-19 deaths each week were also provided by the Hub. Their reference data source is the Center for Systems Science and Engineering (CSSE) at John Hopkins University.

### Dataset

The Hub carries out screening tests for inclusion in their 'ensemble' forecast combinations. Screening excludes forecasts with an incomplete set of quantiles or prediction horizons, and improbable forecasts. The definition of improbable forecasts relate to cumulative deaths, and currently includes decreasing quantiles over the forecast horizons, decreasing cumulative deaths over time (except including an adjustment due to reporting revisions, which is permitted up to a maximum of 10% in the 1-week ahead forecasts), and forecasts of cumulative deaths for a particular location exceeding the size of its population. Before the week of 28 July 2020, the Hub also excluded outlying forecasts, which were identified by a visual check against the actual number of deaths. We only included forecasts that passed the Hub's screening tests.

Our dataset included forecasts projected from forecast origins at midnight on Saturdays between 9 May 2020 to 8 January 2022 for forecasts of cumulative COVID-19 deaths (88 weeks of data), and between 6 June 2020 and 8 January 2022 for forecasts of incident deaths (84 weeks of data). Forecasts of incident deaths were not screened by the Hub in the weeks ending 9 May 2020 to 30 May 2020. We included forecasts of cumulative deaths in this period as we wished to use all the available data, and also given the fact that the set of incident and cumulative forecasts were different in terms of the included models and locations. In terms of the actual weekly COVID-19 mortality, for each location and week, we used the values made available on 15 January 2022. We studied forecasts of COVID-19 mortality for the U.S. as a whole and 51 U.S. jurisdictions, including the 50 states and the District of Columbia. For simplicity, we refer to these as 51 states.

Our analysis included forecasts from 60 forecasting models and the Hub's ensemble model. In the early weeks of our dataset, the majority were susceptible-exposed-infected-removed (SEIR) compartmental models, but as the weeks passed, other model types became more common (Fig 1). These involved methods such as neural networks, agent-based modelling, time series analysis, and the use of curve fitting techniques. S1 Table provides a list of all the models.

Fig 2 shows the extent of missing data for forecasts of incident deaths. The timeline of forecasts from each model (represented by a row) illustrates the extent of missing data across the 52 locations, including the frequent 'entry and exit' of forecasting teams. The corresponding

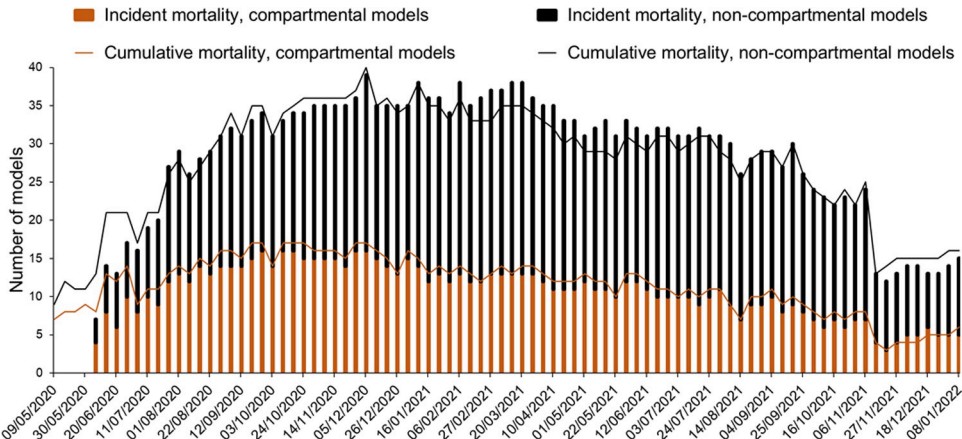

**Fig 1. Number and types of models at each forecast origin for each mortality dataset.**

figure for forecasts of cumulative deaths is given in S1 Fig. Higher levels of forecasts were excluded for cumulative deaths than for incident deaths, and this was mainly attributed to the screening tests, as opposed to exclusion due to not being assessed by the Hub. The extent of missing data was such that imputation was impractical.

Several combining methods required parameter estimation, which we performed for each location and forecast origin. We defined the in-sample estimation period as being initially the first 10 weeks, and then expanding week by week. This resulted in out-of-sample forecasts produced from 78 weekly forecast origins for the cumulative deaths series, and 74 weekly forecast origins for incident deaths.

## Evaluating the interval forecasts

We evaluated out-of-sample prediction accuracy and calibration, with reference to the reported death counts on 15 January 2022, thus producing a retrospective evaluation. Calibration was assessed by the percentage of actual deaths that fell below each bound of the interval forecasts. As each bound is a quantile, this amounted to assessing the calibration of the 23 quantiles for which the teams submitted forecasts. We present this using reliability diagrams. To evaluate prediction accuracy of an interval forecast, we used the interval score (IS) given by

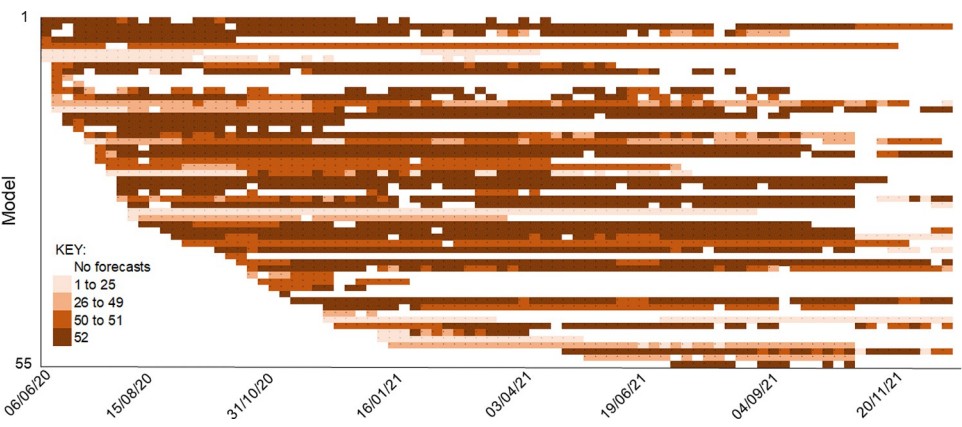

**Fig 2. Data availability for forecasts of incident COVID-19 deaths.**

the following expression [33, 34]:

$$IS_\alpha = (u_t - l_t) + \frac{2}{\alpha}I\{y_t \le l_t\}(l_t - y_t) + \frac{2}{\alpha}I\{y_t \ge u_t\}(y_t - u_t)$$

where $l_t$ is the interval's lower bound, $u_t$ is its upper bound, $y_t$ is the observation in period $t$, $I$ is the indicator function (1 if the condition is true and 0 otherwise), and $\alpha$ is the ideal probability of the observation falling outside the interval. We report the IS for the 95% interval forecasts (for which $\alpha = 5\%$). Lower values of the IS reflect greater interval forecast accuracy. The unit of the IS is deaths. As each forecasting team provides forecasts for 23 different quantiles, the following $K = 11$ symmetric interval forecasts can be considered: 98%, 95%, 90%, 80%, 70%, 60%, 50%, 40%, 30%, 20% and 10%. To summarise prediction accuracy for all these intervals, we used the weighted IS (WIS) [16]:

$$WIS = \frac{1}{K + 1/2} \times \left( w_0 \times 2 \times |y_t - m| + \sum_{k=1}^{K}(w_k \times IS_{\alpha_k}) \right)$$

where $w_0 = \frac{1}{2}$, $w_i = \frac{\alpha_i}{2}$ and $m$ is the forecast of the median. The IS and the WIS are useful for comparing methods, and although their units are deaths, these scores are not interpretable. Averaging each of these two scores across weeks provided the mean IS (MIS) and the mean WIS (MWIS).

We also averaged the scores across forecast horizons. We did this for conciseness, and because we had a relatively short analysis period, which is a particular problem when evaluating forecasts of extreme quantiles. To show the consistency across horizons, we present a set of results by horizon for interval forecasts for both the incident and cumulative deaths data. For this analysis, because we were looking at individual horizons, we were able to use the Diebold-Mariano statistical test [35], adapted to test across multiple series. This test statistic was originally designed to apply to the difference between the mean of an accuracy measure for two methods for a single time series. To compare the difference averaged across multiple time series, we calculated the variance of the sampling distribution by first summing each variance of the sampling distribution from the Diebold-Mariano test applied to each series, and then dividing by the square of the number of series. To summarise results averaged across the four horizons, we were unable to use the adapted Diebold-Mariano test, so we applied the statistical test proposed by Koning et al. [36]. This test compares the rank of each method, averaged across multiple series, with the corresponding average rank of the most accurate method. Statistical testing was based on a 5% significance level.

We present results of the forecast accuracy evaluation in terms of the 95% interval MIS, MWIS, ranks and skill scores, which are calculated as the percentage by which a given method is superior to the mean combination. The mean is a common choice of benchmark in combining studies. We report results for the series of total U.S. deaths, as well as results averaged across all 52 locations. In addition, to avoid scores for some locations dominating, we also present results averaged for three categories, each including 17 states: high, medium and low mortality states. This categorisation was based on the number of cumulative COVID-19 deaths on 15 January 2022. All results are for the out-of-sample period, and to provide some insight into the potential change in ranking of methods over time, we present MWIS results separately for the first and second halves of the out-of-sample period.

We evaluated the effects of changes in reporting patterns on forecast accuracy. Changes in reporting patterns may involve reporting delays of death counts and changes in the definitions of COVID-19 deaths, both of which may lead to backdating of death counts and steep increases or decreases. Backdating of death counts would produce a problematic assessment in

our retrospective evaluation of forecast accuracy, and sudden changes in death counts might cause some forecasting models to misestimate, particularly time series models. To obtain some insight, we compared reports of cumulative death counts for each location in files that were downloaded at multiple time points between 20 June 2020 and 15 January 2022. Locations for which there were notable effects of reporting patterns were excluded in sensitivity analysis. We also examined the effect of outlying forecasts on forecast accuracy by comparing the performance of the mean and median, and visually comparing plots of the MWIS of the mean and median forecasts by location.

Data preparation and descriptive analysis was carried out using Stata version 16 and the forecasts were combined using version 19 of the GAUSS programming language.

## Forecast combining methods

All the interval combining methods are applied to each interval bound separately, and for each mortality series, forecast origin and prediction horizon. The comparison included several interval combining methods that do not rely on the availability of records of past accuracy *for individual models*. These methods include the well-established mean and median combinations [37–39], and the more novel symmetric, exterior, interior and envelope trimming methods, which exclude a particular percentage of forecasts, and then average the remaining forecasts of each bound [40]. Fig 3 provides a visual representation of these methods, which we describe in more detail below.

We also implemented two inverse score methods that do rely on the availability of a record of historical accuracy for each individual model. For any combining method that involved a parameter, such as a trimming parameter, we optimised its value for each location by minimising the MIS calculated over the in-sample period. The following is a list of the combining methods that we included in our study:

a. *Mean* combination. We calculated the average of the forecasts of each bound. This combining method is also known as the simple average.

b. *Median* combination. We calculated the median of the forecasts of each bound. This method is robust to outliers.

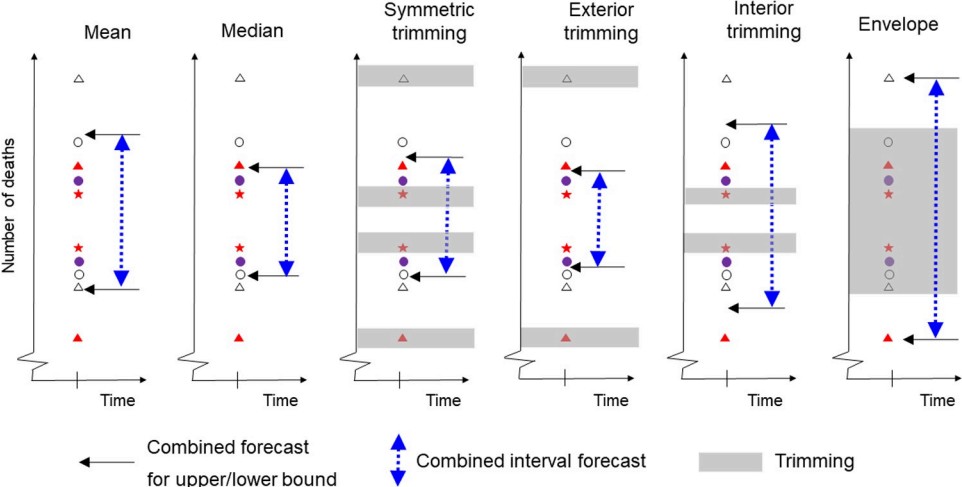

**Fig 3. Illustration of interval forecast combining methods that do not rely on past historical accuracy.** Each pair of shapes represents an interval forecast produced by an individual model.

c. *Ensemble*. This is the COVID-19 Hub ensemble forecast, which was originally the mean combination of the eligible forecasts until the week commencing 28 July 2020, when the ensemble forecast became the median combination and then, in the week commencing 27 September 2021, the Hub switched to using a weighted ensemble method. The use of eligibility screening implies that the ensemble is constructed with the benefit of a degree of trimming which initially involved some subjectivity and was then formalised more objectively. The results for the median and the Hub ensemble will be similar as the latter method was the median combination for around 90% of our out-of-sample period.

d. *Symmetric trimming*. This method deals with outliers. For each bound, it involves trimming the $N$ lowest-valued and $N$ highest-valued forecasts, where $N$ is the largest integer less than or equal to the product of $\beta/2$ and the total number of forecasts, where $\beta$ is a trimming parameter. The median combination is an extreme form of symmetric trimming.

e. *Exterior trimming*. This method targets overly wide intervals. It involves removing the $N$ lowest-valued lower bound forecasts and the $N$ highest-valued upper bound forecasts, where $N$ is the largest integer less than or equal to the product of the trimming parameter $\beta$ and the number of forecasts. When this resulted in a lower bound being above the upper bound, we replaced the two bounds by their average.

f. *Interior trimming*. This method targets overly narrow intervals. It involves removing the $N$ highest-valued lower bound forecasts and the $N$ lowest-valued upper bound forecasts, where $N$ is defined as for exterior trimming.

g. *Envelope method*. The interval is constructed using the lowest-valued lower bound forecast and highest-valued upper bound forecast. This method is an extreme form of interior trimming.

h. *Inverse interval score* method. This is a method that has the model forecasts weighted by historical accuracy, with the weight for each model inversely proportional to the historical MIS for that team [32], which is calculated in the in-sample period. With the shortest in-sample period being 10 weeks, we considered only forecasting teams for which we had forecasts for at least five past forecast origins. Larger numbers led to the elimination of many forecasters for the early weeks in our out-of-sample period. The following expression gives the weight on forecasting model $i$ at forecast origin $t$:

$$w_{it} = \frac{1/MIS_{i,t}}{\sum_{j=1}^{J} 1/MIS_{j,t}}$$

where $MIS_{i,t}$ is the historical MIS computed at forecast origin $t$ from model $i$, and $J$ is the number of forecasting models included in the combination.

i. *Inverse interval score with tuning*. This method has weights inversely proportional to the MIS and a tuning parameter, $\lambda > 0$, to control the influence of the score on the combining weights [32]. The following expression gives the weight on forecasting model $i$ at forecast origin $t$:

$$w_{it} = \frac{\left(1/MIS_{i,t}\right)^{\lambda}}{\sum_{j=1}^{J} \left(1/MIS_{j,t}\right)^{\lambda}}$$

If $\lambda$ is close to zero, the combination reduces to the mean combination, whereas a large value for $\lambda$ leads to the selection of the model with best historical accuracy. The parameter $\lambda$

was optimised using the same expanding in-sample periods, as for the trimming combining methods. Due to the extent of missing forecasts, we pragmatically computed $MIS_{i,t}$ using all available past forecasts, rather than limit the computation to periods for which forecasts from all models were available. For the models for which forecasts were not available for at least 5 past periods, we set $MIS_{i,t}$ to be equal to the mean of $MIS_{i,t}$ for all other models. An alternative approach, which we employed in our earlier study [32], is to omit from the combination any model for which there is only a very short or non-existent history of accuracy available. The disadvantage of this is that it omits potentially useful forecast information, and this was shown by empirical results.

## Comparison with individual models

A comparison of the results of the combining methods with those of individual models is complicated by none of the individual models providing forecasts that passed the Hub's screening for all past periods and locations. We addressed this in two ways. Firstly, we included a *previous best* method, which at each forecast origin and location, selected the interval forecast of the individual model with lowest in-sample MIS. The aim of this is, essentially, to obtain the interval forecasts of the best of the individual models. Secondly, in our results, we also summarise a comparison of the mean and median combinations with individual models for which we had forecasts for at least half the locations and at least half the out-of-sample period. Our inclusion criteria here is rather arbitrary, but the resulting analysis does help us compare the combining methods with the models of the more active individual teams. In this comparison, we excluded the COVID Hub baseline model, as it is only designed to be a comparator point for the models submitted to the Hub and not a true forecast.

## Results

### Forecasting incident deaths

**Main results for incident deaths.**  Table 1 presents the MIS for 95% interval forecasts and MWIS for the 74-week out-of-sample period for incident mortality. Table 2 presents the corresponding mean skill scores, and Table 3 provides the mean ranks and results of the statistical test proposed by Koning et al. [36]. The weighted inverse scores, the ensemble and median

**Table 1. For incident mortality, 95% interval MIS and MWIS.**

| Method | 95% interval MIS | | | | | MWIS | | | | |
|---|---|---|---|---|---|---|---|---|---|---|
| | All | U.S. | High | Med | Low | All | U.S. | High | Med | Low |
| Mean | 779.6 | 9250.7 | 1249.1 | 472.4 | 119.0 | 55.5 | 897.3 | 80.1 | 28.5 | 8.5 |
| Median | 723.0 | 9623.7 | 1050.3 | 488.4 | 106.7 | 51.6 | 914.4 | 68.1 | 27.6 [a] | 8.1 [a] |
| Ensemble | 727.2 | 10303.7 | 1031.5 | 481.3 | 105.3 | 51.5 | 924.8 | 67.5 [a] | 27.6 [a] | 8.1 [a] |
| Sym trim | 764.0 | 9464.8 | 1187.3 | 481.9 | 111.0 | 55.1 | 912.7 | 78.8 | 27.9 | 8.2 |
| Exterior trim | 824.1 | 10435.8 | 1301.6 | 490.2 | 115.0 | 67.2 | 924.7 | 114.0 | 28.8 | 8.4 |
| Interior trim | 767.0 | 9292.5 | 1228.6 | 456.3 | 114.7 | 57.6 | 907.9 | 86.2 | 28.2 | 8.5 |
| Envelope | 3838.1 | 55853.1 | 6752.0 | 1046.7 | 655.8 | 234.0 | 3289.7 | 408.8 | 84.8 | 28.7 |
| Inv score | 690.0 | 8964.2 | 1030.0 | 451.8 [a] | 101.4 [a] | 53.3 | 843.4 | 77.1 | 28.0 | 8.2 |
| Inv score tuning | 656.7 [a] | 8631.1 [a] | 923.7 [a] | 470.2 | 107.1 | 50.0 [a] | 833.2 [a] | 66.8 | 28.9 | 8.4 |
| Previous best | 872.9 | 11428.8 | 1231.0 | 621.8 | 145.0 | 62.0 | 1061.2 | 81.2 | 35.6 | 10.5 |

Lower values are better.

[a] best method in each column.

**Table 2. For incident mortality, skill scores for 95% interval MIS and MWIS.**

| Method | 95% interval MIS | | | | | MWIS | | | | |
|---|---|---|---|---|---|---|---|---|---|---|
| | All | U.S. | High | Med | Low | All | U.S. | High | Med | Low |
| Mean | 0.0 | 0.0 | 0.0 | 0.0 | 0.0 | 0.0 | 0.0 | 0.0 | 0.0 | 0.0 |
| Median | 8.3 | -4.0 | 14.5 | -2.1 | 12.2 | 6.6 | -1.9 | 10.5 | 3.2 | 6.4 |
| Ensemble | 9.6 | -11.4 | 16.4 | -0.5 | 13.1 | 7.0 [a] | -3.1 | 11.4 [a] | 3.5 [a] | 6.5 [a] |
| Sym trim | 7.0 | -2.3 | 12.1 | 0.7 | 8.4 | 3.8 | -1.7 | 3.9 | 2.4 | 5.6 |
| Exterior trim | -3.2 | -12.8 | -4.1 | -4.8 | -0.2 | -1.9 | -3.1 | -6.5 | -1.3 | 1.9 |
| Interior trim | 4.7 | -0.5 | 4.5 | 4.6 | 5.2 | 0.0 | -1.2 | -1.4 | 1.2 | 0.2 |
| Envelope | -222.6 | -503.8 | -239.4 | -161.2 | -265.0 | -252.4 | -266.6 | -302.2 | -212.3 | -247.7 |
| Inv score | 11.7 [a] | 3.1 | 16.8 | 6.0 [a] | 12.5 [a] | 3.8 | 6.0 | 5.6 | 2.6 | 3.2 |
| Inv score tuning | 8.9 | 6.7 [a] | 19.2 [a] | 0.0 | 6.5 | 3.0 | 7.1 [a] | 9.7 | -1.4 | 0.0 |
| Previous best | -20.1 | -23.5 | -8.4 | -34.3 | -18.9 | -19.7 | -18.3 | -8.4 | -24.5 | -27.1 |

Shows percentage change from the mean. Higher values are better.

[a] best method in each column.

combination were the best performing methods. Overall, (for all 52 locations), Table 2 shows that the performance of the inverse score method was almost 12% better than the mean in forecasting the 95% interval and, considering all interval forecasts, the ensemble and median were around 7% better than the mean. Of the trimming methods, symmetric trimming performed best overall, and was quite competitive compared to the leading methods. The 'previous best' was not competitive against most of the combining methods. The worst results were produced by the envelope method. Tables 1 to 3 report results averaged across the four forecast horizons (1 to 4 weeks ahead). We found similar relative performances of the methods when looking at each forecast horizon separately (S2 Table).

**Changes over time in performance for incident deaths.** In Table 4, the MWIS skill scores are shown separately for the first and second halves of the 74-week out-of-sample period. Recalling that the skill scores assess performance relative to the mean combining method, the table shows that this combining method was notably more competitive for the

**Table 3. For incident mortality, average ranks of the 95% interval MIS and MWIS.**

| Method | 95% interval MIS | | | | | MWIS | | | | |
|---|---|---|---|---|---|---|---|---|---|---|
| | All | U.S. | High | Med | Low | All | U.S. | High | Med | Low |
| Mean | 4.7 [b] | 3.0 | 5.0 | 4.4 | 4.7 | 5.1 [b] | 3.0 | 5.4 | 4.8 | 5.2 |
| Median | 5.1 [b] | 6.0 | 5.1 | 5.7 | 4.5 | 3.7 | 6.0 | 4.4 | 3.4 | 3.2 |
| Ensemble | 4.5 [b] | 7.0 | 4.4 | 5.0 | 3.9 | 3.2 | 8.0 | 2.9 | 3.4 | 3.1 [a] |
| Sym trim | 4.7 [b] | 5.0 | 4.8 | 4.6 | 4.8 | 4.4 | 5.0 | 4.7 | 4.4 | 4.1 |
| Exterior trim | 6.8 [b] | 8.0 | 7.2 [b] | 6.5 [b] | 6.7 [b] | 6.5 [b] | 7.0 | 7.1 [b] | 6.6 [b] | 5.8 |
| Interior trim | 4.1 | 4.0 | 4.9 | 3.4 | 3.9 | 5.4 [b] | 4.0 | 6.1 [b] | 4.5 | 5.7 |
| Envelope | 10.0 [b] | 10.0 | 10.0 [b] | 9.9 [b] | 10.0 [b] | 10.0 [b] | 10.0 | 10.0 [b] | 10.0 | 10.0 |
| Inv score | 2.6 [a] | 2.0 | 2.5 [a] | 2.5 [a] | 2.9 [a] | 2.9 [a] | 2.0 | 2.5 [a] | 3.1 [a] | 3.3 |
| Inv score tuning | 4.5 [b] | 1.0 [a] | 3.7 | 4.7 | 5.4 | 5.2 | 1.0 [a] | 4.2 | 6.1 | 5.6 |
| Previous best | 8.0 [b] | 9.0 | 7.5 [b] | 8.3 [b] | 8.1 [b] | 8.5 [b] | 9.0 | 7.9 [b] | 8.7 [b] | 8.9 [b] |

Lower values are better.

[a] best method in each column

[b] significantly worse than the best method, at the 5% significance level.

**Table 4. For incident mortality, skill scores for MWIS calculated separately for the first and second halves of the 74-week out-of-sample period.**

| Method | 1st Half | | | | | 2nd Half | | | | |
|---|---|---|---|---|---|---|---|---|---|---|
| | All | U.S. | High | Med | Low | All | U.S. | High | Med | Low |
| Mean | 0.0 | 0.0 | 0.0 | 0.0 | 0.0 | 0.0 | 0.0 | 0.0 | 0.0 | 0.0 |
| Median | 8.3 | 2.6 | 14.9 | 5.6 [a] | 4.5 | 2.2 | -9.0 | -1.3 | 1.0 | 7.4 [a] |
| Ensemble | 8.6 [a] | 3.3 | 15.0 | 5.6 [a] | 5.1 [a] | 2.9 [a] | -13.0 | 1.3 | 1.2 | 6.8 |
| Sym trim | 4.1 | 1.3 | 5.4 | 4.1 | 3.1 | 2.8 | -6.6 | 0.3 | 1.1 | 7.3 |
| Exterior trim | -4.2 | -4.2 | -8.4 | -2.7 | -1.8 | 2.4 | -1.3 | 1.2 | 0.9 | 5.2 |
| Interior trim | 0.9 | -0.5 | -1.2 | 2.5 | 1.4 | -1.1 | -2.2 | -0.7 | -0.9 | -1.7 |
| Envelope | -245.3 | -238.4 | -338.1 | -212.5 | -201.0 | -233.1 | -311.9 | -200.2 | -215.2 | -285.9 |
| Inv score | 4.1 | 7.8 | 6.7 | 3.7 | 1.7 | 2.7 | 3.1 [a] | 2.4 [a] | 1.5 [a] | 4.1 |
| Inv score tuning | 5.7 | 20.2 [a] | 15.5 [a] | 0.2 | -0.5 | -2.9 | -13.2 | -5.3 | -2.9 | -0.1 |
| Previous best | -13.4 | -3.1 | 1.9 | -21.5 | -23.2 | -30.1 | -41.8 | -36.0 | -28.5 | -25.4 |

Shows percentage change from the mean. Higher values are better.

[a] best method in each column.

second half of the out-of-sample period than for the first half. Comparing the other methods, we see that the same methods that performed particularly well for the first half of the data also were the best methods for the second half. An exception that was the inverse score tuning method that performed worse for the second half, which is perhaps surprising, as one might expect the tuning parameter to be better estimated for the second half, as more data was available for estimation. Inverse score without tuning would appear to be a more robust method for this dataset. The consistently good performance of the median emphasises the importance of robustness.

**Performance by model type for incident deaths.** To evaluate performance by model type, for each category of mortality series (all, U.S, high, medium and low mortality), in Table 5, we tabulate MWIS skills scores for the combining methods applied separately to each of the following three sets of individual models: all models, compartmental models only, and non-compartmental models only. For each category of mortality series, to enable a comparison of the combining methods applied to the different sets of individual models, we computed the

**Table 5. For incident mortality, skill scores for MWIS for combining methods applied to forecasts of all models, compartmental models only, and non-compartmental models only.**

| Method | All | | | U.S. | | | High | | | Med | | | Low | | |
|---|---|---|---|---|---|---|---|---|---|---|---|---|---|---|---|
| | All | Comp | Non-comp | All | Comp | Non-comp | All | Comp | Non-comp | All | Comp | Non-comp | All | Comp | Non-comp |
| Mean | 0.0 | 5.1 | -8.0 | 0.0 | -15.7 | -3.2 | 0.0 | 10.4 | -12.2 | 0.0 | 1.6 | -6.3 | 0.0 | 4.1 | -6.1 |
| Median | 6.6 | 4.2 | 3.7 | -1.9 | -13.7 | -2.3 | 10.5 | 7.7 | 7.0 | 3.2 | 1.6 | 0.2 | 6.4 | 4.1 | 4.1 |
| Sym trim | 3.8 | 4.7 | 1.6 | -1.7 | -16.0 | -2.2 | 3.9 | 8.9 | 1.9 | 2.4 | 2.2 | -0.8 | 5.6 | 3.9 | 3.7 |
| Exterior trim | -1.9 | 3.5 | -6.4 | -3.1 | -17.7 | -3.4 | -6.5 | 8.7 | -11.0 | -1.3 | 0.3 | -6.6 | 1.9 | 2.4 | -1.8 |
| Interior trim | 0.0 | 5.6 | -6.8 | -1.2 | -15.2 | -4.1 | -1.4 | 10.4 | -11.0 | 1.2 | 2.6 | -4.3 | 0.2 | 4.7 | -5.5 |
| Envelope | -252.4 | -89.6 | -240.0 | -266.6 | -144.8 | -234.6 | -302.2 | -83.9 | -288.4 | -212.3 | -101.3 | -193.8 | -247.7 | -81.5 | -244.7 |
| Inv score | 3.8 | 7.2 [a] | -2.8 | 6.0 | -3.7 | 2.8 | 5.6 | 12.6 [a] | -4.1 | 2.6 | 3.9 [a] | -3.3 | 3.2 | 5.4 | -1.2 |
| Inv score tun | 3.0 | 5.1 | -2.7 | 7.1 [a] | 7.1 [a] | 2.8 | 9.7 | 11.5 | 0.5 | -1.4 | 1.1 | -4.9 | 0.0 | 2.3 | -4.3 |
| Previous best | -19.7 | -13.0 | -21.4 | -18.3 | -9.5 | -7.1 | -8.4 | -9.1 | -7.8 | -24.5 | -20.8 | -193.1 | -27.1 | -9.7 | -29.6 |

Shows percentage change from the mean. Higher values are better.

[a] best method in each of the five mortality categories.

**Table 6. For incident mortality, summary statistics of skill scores for individual models.**

|  | 95% interval MIS | | | | | MWIS | | | | |
|---|---|---|---|---|---|---|---|---|---|---|
| Method | All | U.S. | High | Med | Low | All | U.S. | High | Med | Low |
| *Mean combining as skill score benchmark* | | | | | | | | | | |
| Count | 27 | 26 | 27 | 27 | 27 | 27 | 26 | 27 | 27 | 27 |
| Mean | -117.5 | -251.5 | -107.7 | -121.2 | -116.2 | -96.1 | -160.8 | -88.3 | -97.8 | -105.1 |
| Median | -78.1 | -176.9 | -56.3 | -77.2 | -84.1 | -89.5 | -122.7 | -76.3 | -90.0 | -76.9 |
| Minimum | -486.6 | -937.6 | -439.3 | -426.8 | -542.1 | -263.4 | -651.2 | -353.5 | -253.1 | -506.2 |
| Maximum | -0.8 | -2.2 | 16.8 | -12.3 | 0.6 | -38.9 | -13.6 | -31.9 | -43.7 | -36.3 |
| Number > 0 | 0 | 0 | 2 | 0 | 1 | 0 | 0 | 0 | 0 | 0 |
| *Median combining as skill score benchmark* | | | | | | | | | | |
| Count | 27 | 26 | 27 | 27 | 27 | 27 | 26 | 27 | 27 | 27 |
| Mean | -140.6 | -240.6 | -149.2 | -120.6 | -146.0 | -112.1 | -157.9 | -114.3 | -106.5 | -119.3 |
| Median | -94.2 | -162.3 | -102.0 | -75.4 | -104.0 | -104.2 | -123.2 | -99.5 | -100.9 | -89.0 |
| Minimum | -618.2 | -944.4 | -597.2 | -452.7 | -700.5 | -291.5 | -651.5 | -420.9 | -272.5 | -538.5 |
| Maximum | -11.6 | 3.0 | 0.3 | -12.2 | -13.5 | -49.0 | -11.8 | -50.4 | -48.5 | -39.7 |
| Number > 0 | 0 | 1 | 1 | 0 | 0 | 0 | 0 | 0 | 0 | 0 |

Higher values of the skill score are better.

skill scores using the same benchmark, which we set as the mean combination *of all models*. Note that we have omitted the ensemble from Table 5 because the forecasts from this method were determined by the Hub, and so we were not in control of which individual methods that method combines. The first point to note from Table 5 is that combining only non-compartmental models led to poorer results for almost all combining methods and categories of mortality series. A second point to note is that, for the all, high, medium and low categories of series, combining only compartmental models was preferable to combining all models, unless the combining method was the median. For the median, combining all available models was preferable. It is interesting to note that the two inverse score methods, when applied only to the compartmental models, become competitive with the median.

**Performance of individual models for incident deaths.** Table 6 reports the performance of the 27 individual models for which we had forecasts of incident deaths for at least half the out-of-sample period and at least half of the 52 locations. The table summarises skill scores based on scores calculated for the individual model and the benchmark method using only those weeks for which forecasts were available for the individual model. Table 6 reports results for the skill score calculated using mean combining as the benchmark, as in our previous tables, but also the results for skill score calculate using median combining as the benchmark method. The skill scores of these individual models were highly variable, and generally negative, implying that they were not competitive against the mean or median in any category. The only notable exception was the performance of an individual model that was almost 17% better than the mean for the 95% interval forecasts in the high mortality locations.

**Calibration results for incident deaths.** As we stated earlier, with each bound of the interval forecasts being a quantile, we assess calibration for each of the 23 quantiles for which the teams submitted forecasts. We do this in Fig 4, which presents reliability diagrams for each category of mortality series for the mean, median and inverse score with tuning combining methods. Reasonable calibration can be seen in the plot relating to all 52 locations, and there is good calibration at the extreme quantiles in each plot, except the one for low mortality locations. Most methods had calibration that was too low for the U.S and high mortality locations,

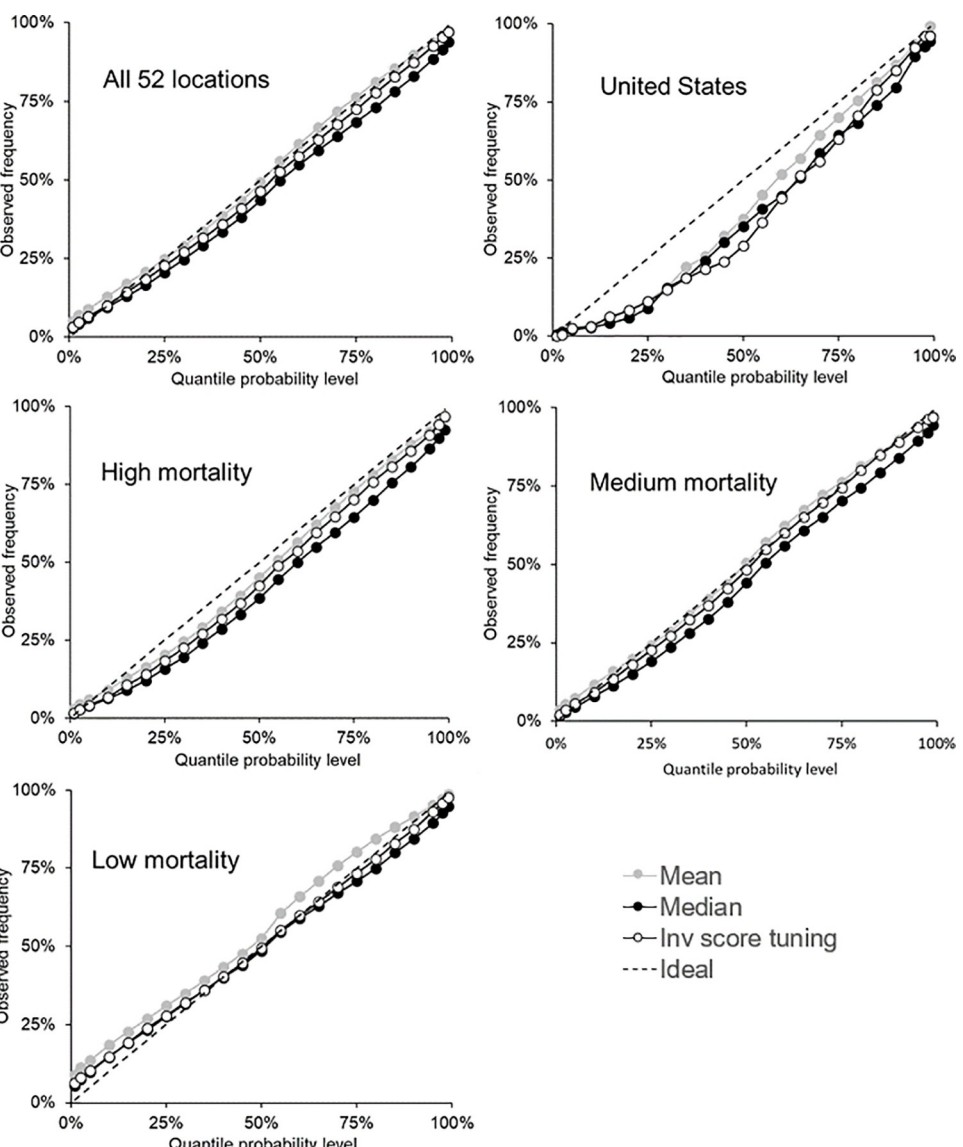

**Fig 4. For incident mortality, reliability diagrams showing calibration of the 23 quantiles for the mean, median and inverse score with tuning methods.** The 23 quantiles include all bounds on the interval forecasts and the median.

and most methods displayed calibration that was too high for the low mortality locations, particularly for the lower quantiles. For the medium mortality locations, the mean and inverse score with tuning performed better than the median, for which calibration was slightly too low. For all methods, S3–S7 Tables provide the calibration for the five categories of the locations: all 52 locations, U.S, high mortality, medium mortality and low mortality, respectively.

## Forecasting cumulative deaths

In this section, for the cumulative deaths data, we report analogous results tables and figure to those that we have presented for the incident deaths data.

**Main results for cumulative deaths.** Table 7 presents the MIS for 95% interval forecasts and MWIS for the 78-week out-of-sample period for cumulative mortality. The corresponding

**Table 7. For cumulative mortality, 95% interval MIS and MWIS for the 78-week out-of-sample period.**

| Method | 95% interval MIS | | | | | MWIS | | | | |
|---|---|---|---|---|---|---|---|---|---|---|
| | All | U.S. | High | Med | Low | All | U.S. | High | Med | Low |
| Mean | 5540 | 87322 | 8822 | 2173 | 814 | 234 | 4156 | 346 | 92 | 33 |
| Median | 2784 [a] | 30188 | 5514 | 921 [a] | 306 [a] | 143 [a] | 2332 | 228 | 54 [a] | 18 [a] |
| Ensemble | 2821 | 33147 | 5445 [a] | 930 | 306 [a] | 143 [a] | 2355 | 226 [a] | 54 [a] | 18 [a] |
| Sym trim | 3044 | 34038 | 5882 | 1066 | 360 | 151 | 2415 | 242 | 57 | 19 |
| Exterior trim | 5388 | 83865 | 8632 | 2127 | 788 | 207 | 3442 | 319 | 82 | 29 |
| Interior trim | 3203 | 30333 | 6293 | 1221 | 501 | 226 | 2933 | 419 | 73 | 27 |
| Envelope | 7915 | 154350 | 10814 | 2931 | 1385 | 1027 | 21143 | 1373 | 402 | 122 |
| Inv score | 3429 | 29134 | 6560 | 1598 | 617 | 170 | 2496 | 277 | 70 | 27 |
| Inv score tuning | 3626 | 24624 [a] | 6818 | 2237 | 589 | 161 | 2270 [a] | 259 | 73 | 25 |
| Previous best | 4786 | 33090 | 8593 | 3632 | 467 | 180 | 2580 | 281 | 94 | 24 |

Lower values are better.

[a] best method in each column.

skill scores are in Table 8. The median and ensemble approaches were the best performing methods in terms of both metrics. Overall, their performance for the 95% interval was about 65% better than the mean, and considering all interval forecasts, the ensemble and median were 43% better than the mean. The very poor performance of the mean suggests the presence of outlying forecasts. These would also undermine the weighted inverse score methods, as they involve weighted averages. The inverse score with tuning method was the best method for the U.S. series. Interior trimming performed better than the inverse scoring methods for the 95% interval, which suggests that there were large numbers of 95% intervals that were too narrow. Both metrics showed symmetric trimming performing almost as well as the median. Table 9 reports mean ranks. Using the statistical test proposed by Koning et al. [36], we identified that, in terms of the mean rank, most methods were statistically significantly worse than the median-based approaches. We found similar relative performances of the methods when looking at each forecast horizon separately (S8 Table).

**Table 8. For cumulative mortality, skill scores for 95% interval MIS and MWIS for the 78-week out-of-sample period.**

| Method | 95% interval MIS | | | | | MWIS | | | | |
|---|---|---|---|---|---|---|---|---|---|---|
| | All | U.S. | High | Med | Low | All | U.S. | High | Med | Low |
| Mean | 0.0 | 0.0 | 0.0 | 0.0 | 0.0 | 0.0 | 0.0 | 0.0 | 0.0 | 0.0 |
| Median | 65.2 [a] | 65.4 | 60.7 [a] | 63.6 [a] | 70.5 [a] | 43.2 [a] | 43.9 | 39.7 | 41.7 [a] | 47.9 [a] |
| Ensemble | 64.9 | 62.0 | 60.5 | 63.4 | 70.2 | 43.1 | 43.3 | 40.0 [a] | 41.6 | 47.4 |
| Sym trim | 60.4 | 61.0 | 55.1 | 59.0 | 66.1 | 39.9 | 41.9 | 36.6 | 38.0 | 44.6 |
| Exterior trim | 4.7 | 4.0 | 4.3 | 2.8 | 7.0 | 11.2 | 17.2 | 10.6 | 10.1 | 12.5 |
| Interior trim | 46.3 | 65.3 | 45.6 | 47.0 | 45.0 | 17.2 | 29.4 | 15.6 | 19.6 | 15.4 |
| Envelope | -46.2 | -76.8 | -36.5 | -47.7 | -53.2 | -318.1 | -408.8 | -314.9 | -349.5 | -287.3 |
| Inv score | 32.4 | 66.6 | 37.0 | 28.3 | 28.7 | 23.3 | 39.9 | 26.3 | 22.9 | 19.5 |
| Inv score tuning | 22.5 | 71.8 [a] | 27.3 | 2.0 | 30.6 | 23.5 | 45.4 [a] | 26.8 | 19.5 | 22.5 |
| Previous best | 25.2 | 62.1 | 18.4 | -18.1 | 54.7 | 17.5 | 37.9 | 16.1 | 4.5 | 28.8 |

Shows percentage change from the mean. Higher values are better.

[a] best method in each column.

**Table 9. For cumulative mortality, average ranks of the 95% interval MIS and MWIS.**

| Method | 95% interval MIS | | | | | MWIS | | | | |
|---|---|---|---|---|---|---|---|---|---|---|
| | All | U.S. | High | Med | Low | All | U.S. | High | Med | Low |
| Mean | 7.8 [b] | 9.0 | 8.1 [b] | 7.3 [b] | 8.1 [b] | 8.4 [b] | 9.0 | 8.4 [b] | 8.5 [b] | 8.4 [b] |
| Median | 2.4 [a] | 3.0 | 2.8 [a] | 2.0 [a] | 2.5 [a] | 1.9 | 2.0 | 2.1 | 1.6 | 1.9 [a] |
| Ensemble | 2.6 | 6.0 | 2.8 [b] | 2.1 | 2.7 | 1.8 [a] | 3.0 | 1.8 [a] | 1.4 [a] | 2.1 |
| Sym trim | 3.8 | 7.0 | 3.7 | 3.8 | 3.8 | 3.2 [b] | 4.0 | 3.1 | 3.1 | 3.2 |
| Exterior trim | 7.7 [b] | 8.0 | 7.5 [b] | 7.5 [b] | 8.0 [b] | 7.1 [b] | 8.0 | 7.4 [b] | 6.8 [b] | 6.9 [b] |
| Interior trim | 4.0 | 4.0 | 4.5 | 3.2 | 4.4 | 6.0 [b] | 7.0 | 6.3 [b] | 5.6 [b] | 6.2 [b] |
| Envelope | 8.9 [b] | 10.0 | 9.1 [b] | 9.1 [b] | 8.5 [b] | 10.0 [b] | 10.0 | 10.0 [b] | 10.0 [b] | 10.0 [b] |
| Inv score | 5.4 [b] | 2.0 | 5.2 | 5.2 | 6.1 [b] | 5.1 [b] | 5.0 | 5.0 [b] | 4.6 [b] | 5.8 [b] |
| Inv score tuning | 6.2 [b] | 1.0 [a] | 5.5 | 7.2 [b] | 6.1 [b] | 5.1 [b] | 1.0 [a] | 4.5 [b] | 5.7 [b] | 5.4 [b] |
| Previous best | 6.0 [b] | 5.0 | 5.7 | 7.6 [b] | 4.8 | 6.4 [b] | 6.0 | 6.5 [b] | 7.6 [b] | 5.2 [b] |

Lower values are better.

[a] indicates best method in each column

[b] significantly worse than the best method, at the 5% significance level.

**Changes over time in performance for cumulative deaths.** The skill scores for the MWIS for the first and second halves of the 74-week out-of-sample period are shown in Table 10. For the first half of the out-of-sample period, the improvements over the mean were considerably smaller than for the second half. The sizeable skill scores for the second half for the ensemble, median and symmetric trimming strongly suggests the presence of outliers. We consider this issue further in a later section where we investigate the impact of reporting patterns and outliers on forecast accuracy. We also note in Table 10 that the inverse score methods were more competitive against the ensemble and median in the first half of the out-of-sample period.

**Performance by model type for cumulative deaths.** The MWIS results of the comparison by model type for cumulative forecasts are reported in Table 11. For all combining methods except the median, combining only compartmental models performed better than combining all models for all categories of the mortality series, except the category that is just the total U.S.

**Table 10. For cumulative mortality, skill scores for MWIS calculated separately for the first and second halves of the 78-week out-of-sample period.**

| Method | 1st Half | | | | | 2nd Half | | | | |
|---|---|---|---|---|---|---|---|---|---|---|
| | All | U.S. | High | Med | Low | All | U.S. | High | Med | Low |
| Mean | 0.0 | 0.0 | 0.0 | 0.0 | 0.0 | 0.0 | 0.0 | 0.0 | 0.0 | 0.0 |
| Median | 8.9 | 6.9 | 8.5 | 8.2 | 10.2 | 65.7 [a] | 68.3 [a] | 62.7 | 64.8 [a] | 69.2 [a] |
| Ensemble | 9.1 [a] | 7.0 | 8.7 | 8.3 [a] | 10.4 [a] | 65.6 | 67.4 | 63.5 [a] | 64.6 | 68.2 |
| Sym trim | 8.4 | 4.3 | 8.1 | 6.9 | 10.5 | 60.4 | 66.9 | 58.1 | 60.1 | 62.6 |
| Exterior trim | -0.1 | -0.4 | 0.1 | -1.2 | 0.8 | 18.4 | 29.1 | 18.4 | 18.1 | 18.2 |
| Interior trim | 0.7 | 5.3 | -4.4 | 3.0 | 3.0 | 28.8 | 45.5 | 31.9 | 30.8 | 22.1 |
| Envelope | -240.0 | -195.0 | -246.1 | -246.9 | -230.2 | -367.6 | -555.0 | -357.8 | -419.8 | -321.2 |
| Inv score | 7.6 | 10.8 | 9.3 | 5.9 | 7.2 | 33.9 | 59.2 | 37.5 | 35.6 | 26.2 |
| Inv score tuning | 8.4 | 17.0 [a] | 11.2 [a] | 5.1 | 8.4 | 35.0 | 64.0 | 38.5 | 32.4 | 31.4 |
| Previous best | -9.5 | 1.0 | -6.5 | -17.2 | -5.8 | 38.8 | 62.2 | 34.2 | 28.2 | 50.1 |

Shows percentage change from the mean. Higher values are better.

[a] best method in each column.

**Table 11. For cumulative mortality, skill scores for MWIS for combining methods applied to forecasts of all models, compartmental models only, and non-compartmental models only.**

| Method | All | All Comp | All Non-comp | U.S. All | U.S. Comp | U.S. Non-comp | High All | High Comp | High Non-comp | Med All | Med Comp | Med Non-comp | Low All | Low Comp | Low Non-comp |
|---|---|---|---|---|---|---|---|---|---|---|---|---|---|---|---|
| Mean | 0.0 | 39.7 | -27.3 | 0.0 | 33.1 | -27.1 | 0.0 | 36.8 | -29.0 | 0.0 | 37.3 | -27.8 | 0.0 | 45.1 | -25.2 |
| Median | 43.2 | 40.7 | 41.1 | 43.9 | 33.2 | 45.0 | 39.7 | 37.3 | 36.7 | 41.7 | 39.1 | 39.3 | 47.9 | 45.7 | 46.5 |
| Sym trim | 39.9 | 40.8 | 38.9 | 41.9 | 32.6 | 43.0 | 36.6 | 37.4 | 33.9 | 38.0 | 38.9 | 37.3 | 44.6 | 46.0 | 44.8 |
| Exterior trim | 11.2 | 38.7 | -4.8 | 17.2 | 31.3 | -0.2 | 10.6 | 35.9 | -7.7 | 10.1 | 36.0 | -5.6 | 12.5 | 44.4 | -1.5 |
| Interior trim | 17.2 | 40.1 | 9.7 | 29.4 | 35.1 | 16.5 | 15.6 | 36.5 | 4.8 | 19.6 | 38.0 | 12.2 | 15.4 | 45.6 | 11.5 |
| Envelope | -318.1 | -27.3 | -337.3 | -408.8 | -14.5 | -423.1 | -314.9 | -29.0 | -324.0 | -349.5 | -43.8 | -354.5 | -287.3 | -12.0 | -329.5 |
| Inv score | 23.3 | 40.9 | 14.0 | 39.9 | 38.1 | 35.9 | 26.3 | 37.4 | 16.0 | 22.9 | 39.1 | 12.1 | 19.5 | 46.1 | 12.2 |
| Inv score tun | 23.5 | 40.1 | 15.7 | 45.4 | 44.2 | 39.2 | 26.8 | 37.1 | 22.0 | 19.5 | 37.9 | 8.2 | 22.5 | 44.6 | 14.8 |
| Previous best | 17.5 | 29.0 | 7.8 | 37.9 | 37.1 | 40.2 | 16.1 | 24.4 | 16.1 | 4.5 | 25.1 | -1.1 | 28.8 | 36.5 | 5.3 |

Shows percentage change from the mean. Higher values are better.

deaths. As with forecasts of incident deaths, the inverse score methods were competitive with the median when combining forecasts only from compartmental models.

**Performance of individual models for cumulative deaths.** For forecasts of cumulative mortality, 25 models provided forecasts for at least half the locations for at least half the weeks in the out-of-sample period. As can be seen in Table 12, the performance of these individual models was highly variable. The upper half of the table shows that, particularly for the 95% interval, a good number of the individual models were able to outperform the mean. However, the lower half of the table shows that the individual methods were not competitive with the median, except for the case of the 95% interval for the total U.S. mortality series.

**Calibration results for cumulative deaths.** Fig 5 presents reliability diagrams for each category of the mortality series to summarise the calibration for each of the 23 quantiles for forecasts of the mean, median and inverse score with tuning combining methods. The figure

**Table 12. For cumulative mortality, summary statistics of skill scores for individual models, using mean and median combining as benchmark.**

| Method | 95% interval MIS All | 95% interval MIS U.S. | 95% interval MIS High | 95% interval MIS Med | 95% interval MIS Low | MWIS All | MWIS U.S. | MWIS High | MWIS Med | MWIS Low |
|---|---|---|---|---|---|---|---|---|---|---|
| *Mean combining as skill score benchmark* | | | | | | | | | | |
| Count | 25 | 24 | 25 | 25 | 25 | 25 | 24 | 25 | 25 | 25 |
| Mean | -17.9 | -9.4 | -14.7 | -19.2 | -24.2 | -43.0 | -21.1 | -34.6 | -46.2 | -56.4 |
| Median | 2.7 | 19.6 | 15.4 | 8.0 | -14.2 | -36.3 | -4.7 | -23.1 | -39.3 | -37.0 |
| Minimum | -255.2 | -307.7 | -224.5 | -261.1 | -245.6 | -131.2 | -166.8 | -113.2 | -129.0 | -390.8 |
| Maximum | 61.2 | 74.6 | 57.7 | 55.4 | 69.5 | 18.6 | 41.7 | 16.8 | 14.7 | 31.0 |
| Number > 0 | 13 | 17 | 15 | 14 | 10 | 4 | 11 | 3 | 4 | 6 |
| *Median combining as skill score benchmark* | | | | | | | | | | |
| Count | 25 | 24 | 25 | 25 | 25 | 25 | 24 | 25 | 25 | 25 |
| Mean | -165.1 | -136.4 | -155.4 | -168.0 | -181.7 | -126.8 | -108.8 | -112.3 | -132.7 | -145.3 |
| Median | -142.6 | -130.4 | -108.6 | -137.9 | -148.1 | -107.4 | -87.3 | -93.9 | -115.6 | -102.2 |
| Minimum | -573.6 | -480.9 | -481.6 | -511.7 | -991.6 | -310.6 | -371.8 | -228.2 | -287.2 | -616.1 |
| Maximum | -14.6 | 20.3 | -6.0 | -26.2 | -6.3 | -43.0 | -7.4 | -41.7 | -45.8 | -41.6 |
| Number > 0 | 0 | 2 | 0 | 0 | 0 | 0 | 0 | 0 | 0 | 0 |

Higher values of the skill score are better.

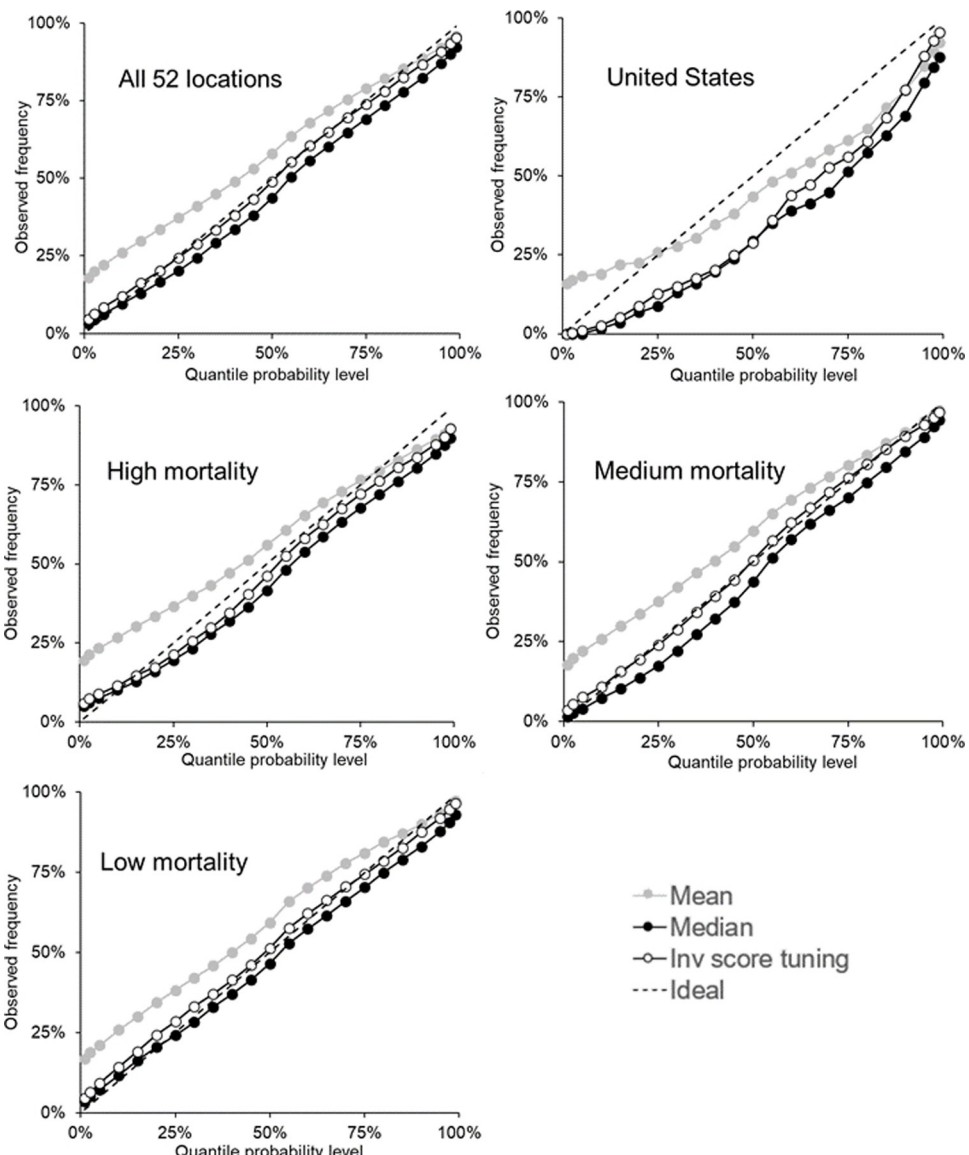

**Fig 5. For cumulative mortality, reliability diagrams showing calibration of the 23 quantiles for the mean, median and inverse score with tuning methods.** The 23 quantiles include all bounds on the interval forecasts and the median.

shows that the mean produced quantile forecasts that tended to be too low for the U.S. series, and too high for the other four categories. The inverse score with tuning method was very well calibrated, except for the U.S. series, and the median method also performed reasonably well, although it tended to produce quantile forecasts that were generally a little low. For all methods, S9–S13 Tables show the calibration for all methods for the five categories of the locations: all 52 locations, U.S, high mortality, medium mortality and low mortality, respectively.

## Impact of reporting patterns and outliers on forecast accuracy

We observed changes in reporting patterns of historical death counts in 15 locations. Fig 6 shows examples of six locations where updates to death counts were particularly notable. We found evidence of backdating in Delaware, Ohio, Rhode Island and Indiana. Backdating of

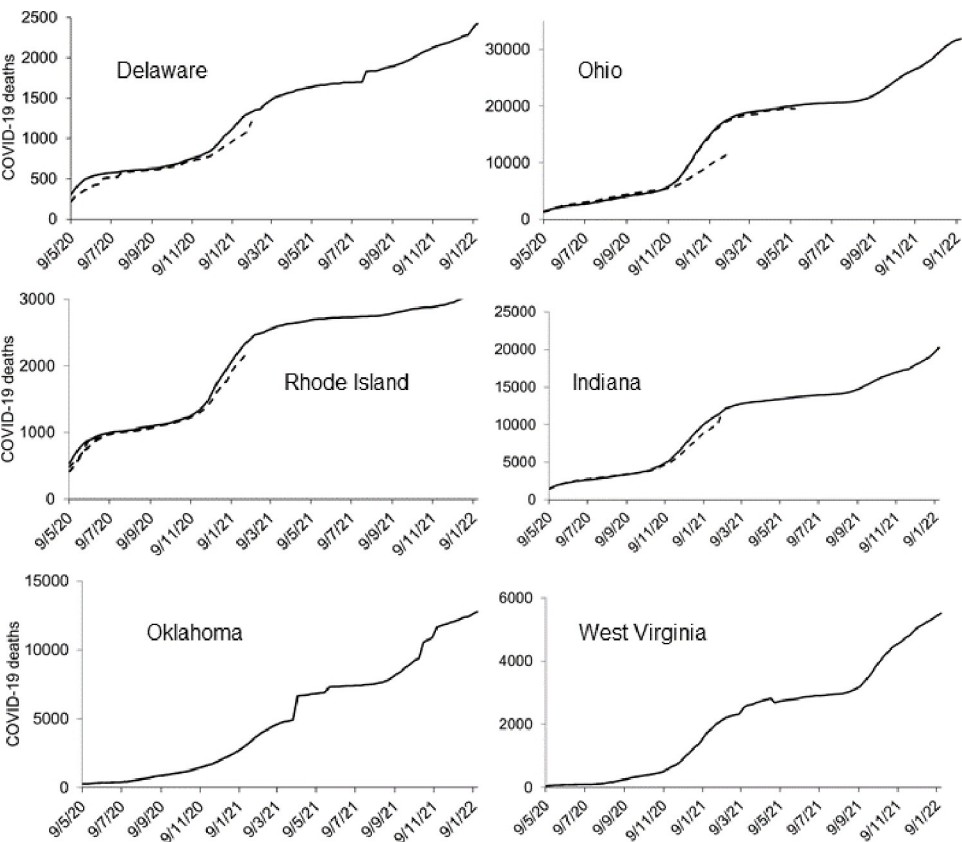

**Fig 6. Numbers of reported cumulative deaths in six states where there were noticeable changes in reporting patterns.** Based on reported death counts at multiple data points between 20 June 2020 and 15 January 2022.

historical death counts is shown as dashed lines. We noted a sharp drop in death counts in West Virginia in May 2021, suggesting a redefinition of COVID-19 deaths. There were sharp increases in death counts in Oklahoma in early April 2021 and in Delaware in late July 2021. We also observed sharp increases in death counts of two other locations (Missouri and Nebraska).

For each of the 51 states, Figs 7 and 8 present the MWIS for the mean, median and inverse score with tuning method for incident and cumulative mortalities, respectively. The locations are ordered by the cumulative number of deaths on 15 January 2022. In both figures, all three methods performed noticeably poorly for Ohio, Oklahoma, Nebraska and West Virginia, for which we found notable changes in reporting patterns, as well as in Virginia and Oregon, where we did not observe such changes. We cannot rule out there having been changes in reporting patterns for these and other locations, as we did not have a complete set of files of reported death counts for each week.

As a sensitivity analysis, we excluded the eight named locations for which there were noticeable changes in reporting patterns. The resulting MWIS skill scores are given in S14 Table and S15 Table for incident and cumulative deaths respectively. Compared to the MWIS skill scores presented in Tables 2 and 8 (where no locations were excluded), there were improvements for all methods, slight changes in rankings, but no changes in the overall conclusions.

The differences between the performance of the mean and median forecasts described in previous sections and highlighted Fig 8 suggested a problem with outliers, particularly for cumulative deaths. Fig 9 provides some insight into an outlying set of 23 quantile forecasts.

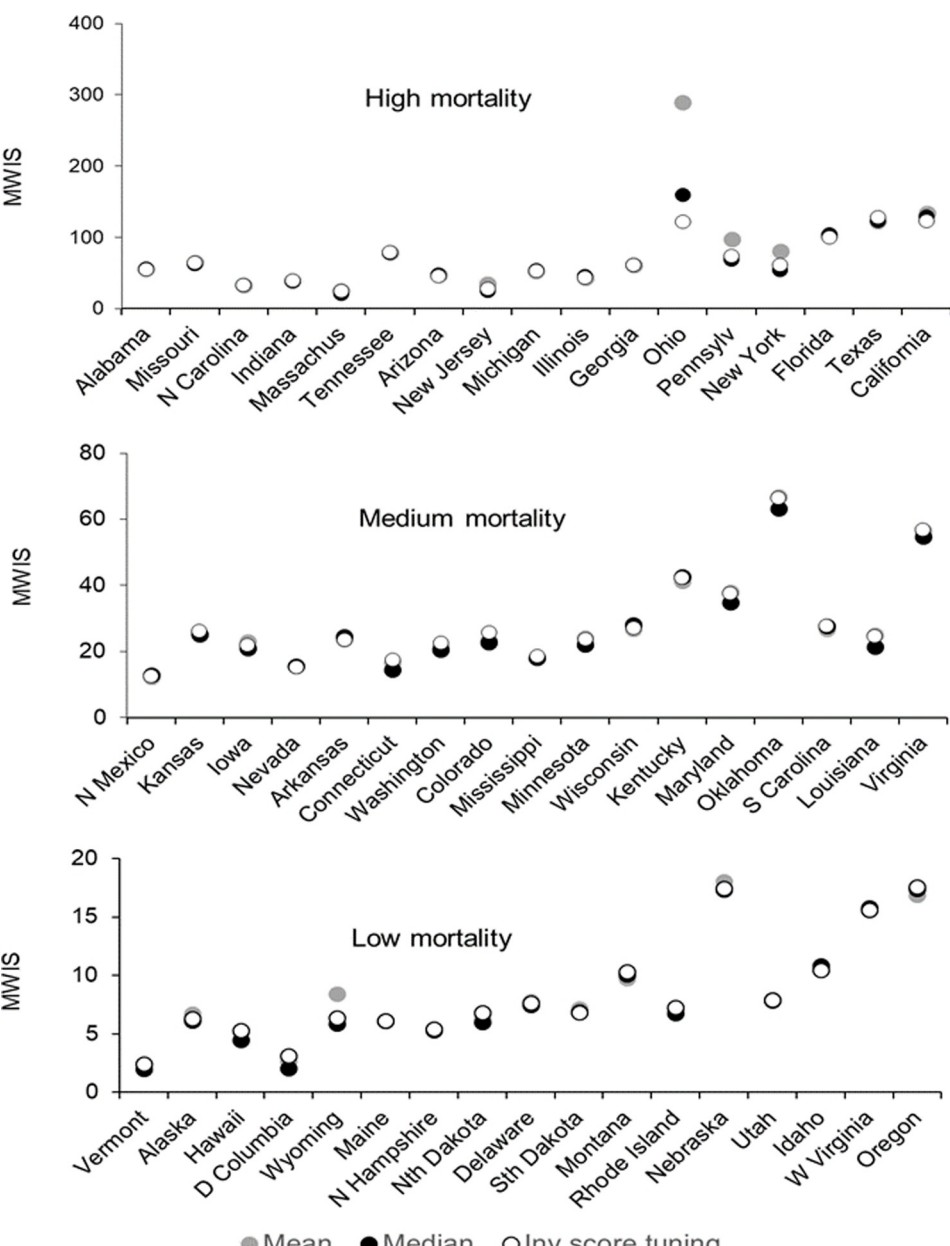

**Fig 7. For incident mortality, MWIS for high, medium and low mortality states for three selected combining methods.**

Each line shows the probability distributions function mapped out by the 23 quantile forecasts of an individual model. For each of the two locations, the presence of an outlying set of quantile forecasts is evident by there being a line that differs notably from the other lines.

## Discussion

The weighted inverse scores, ensemble and median performed best for forecasts of incident deaths. They produced moderate improvements in performance over the common benchmark mean combination. With the forecasts of cumulative deaths, improvements over the mean

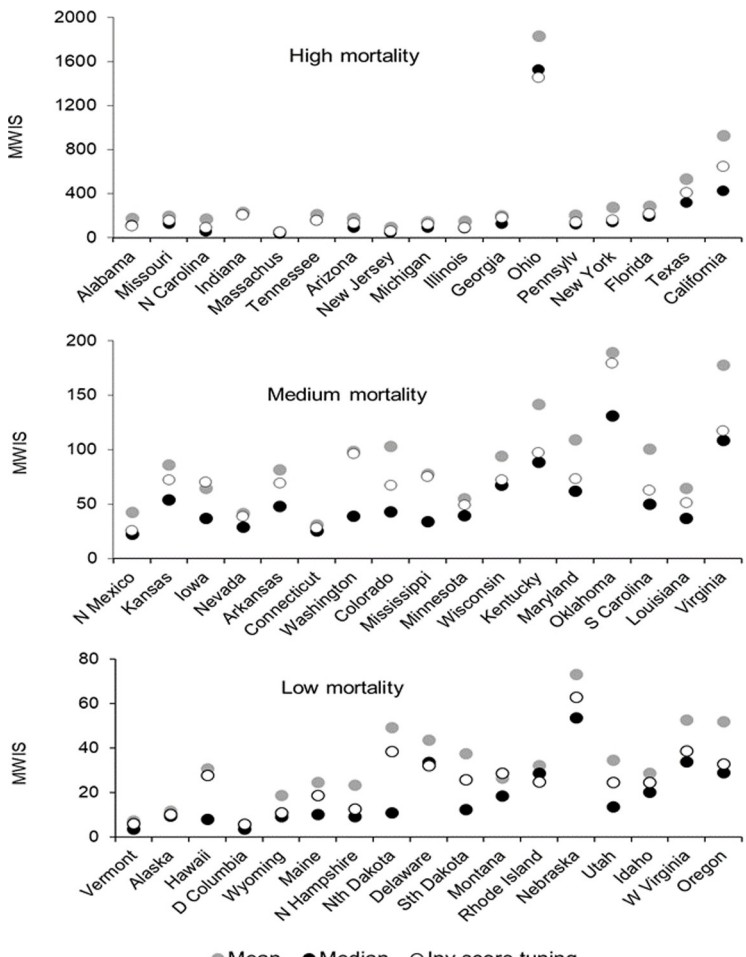

**Fig 8. For cumulative mortality, MWIS for high, medium and low mortality states for three selected combining methods.**

were much higher, and for the median and ensemble, they were substantial. For all combining methods except the median, combining forecasts from only compartmental models produced better forecasts than forecasts from combinations of all models. Furthermore, considering combinations of compartmental models only, inverse score combining was more competitive against the median for both mortalities. We found that the individual models were not competitive with the better combining methods. The presence of outlying forecasts had an adverse impact on the performance of the mean and the inverse score methods, which involved weighted averaging. The adverse effects of reporting patterns on performance were minor.

We presented the inverse score methods in an earlier study of forecasts of cumulative COVID-19 deaths [32]. The current paper considers both incident and cumulative forecasts, using a far longer period of data than the earlier study, and involves a different set of forecasting models, as we now only include forecasts that passed the screening tests of the COVID-19 Hub. In our earlier study, the inverse score methods were the most accurate overall and the mean generally outperformed the median. The mean was also competitive against the inverse score method for many locations. The results of the current study for forecasts of cumulative deaths were not consistent with those of the earlier study, although much better results were achieved for the mean by combining only compartmental models, and when combining

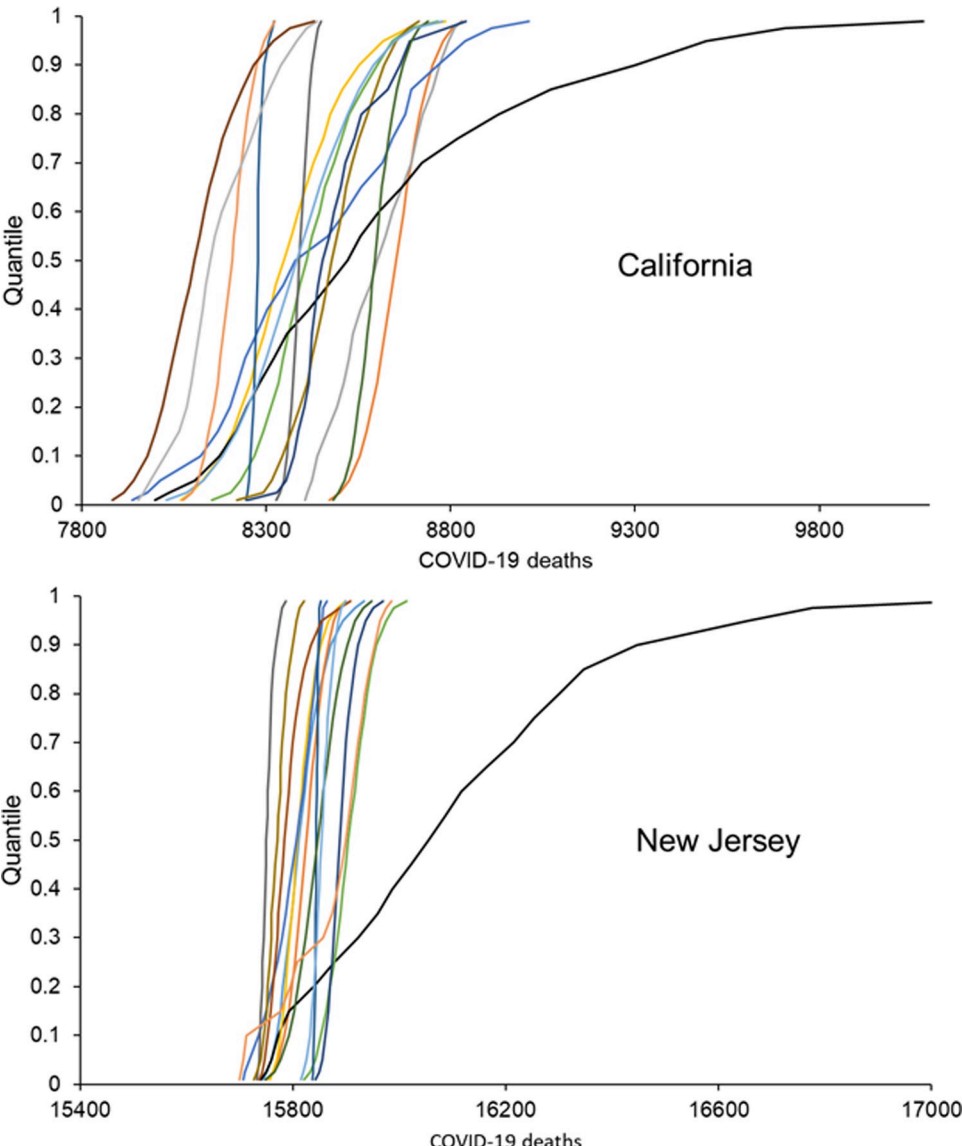

**Fig 9. Two examples of an outlying set of quantile forecasts of cumulative deaths for one week-ahead from forecast origin for the week ending on 18 July 2020.**

forecasts of incident deaths from all models, the relative performance of the inverse mean methods was considerably better. In the current study of cumulative deaths, the leading methods were generally the ensemble, the median and symmetric mean (for which the median is an extreme case). These methods are robust to outliers. The results of our two studies illustrate that, particularly for forecasts of cumulative deaths, the relative performance of combining methods depends on the extent of outlying forecasts, and that outlying forecasts were clearly more prevalent in the dataset for the current study.

Another relevant previous study is that by Bracher et al. [29], who compared forecasts produced by the mean combination, median combination and a weighted combination for COVID-19 deaths in Germany and Poland. They found that combined methods did not perform better than individual models. However, this study was limited by an evaluation period

of only ten weeks. It is also worth noting that the study used just thirteen individual models in the combinations. In our previous work [32], we found accuracy improved notably as the number of individual models rose, plateauing at around twenty models.

Previous studies have found that data driven models can perform better than compartmental models in forecasting COVID-19 data [9, 41, 42]. For forecasts of both mortalities, in many cases we found that there was no benefit in including non-compartmental models in a combination with compartmental models for a number of combining methods, including the mean combination. Non-compartmental models include simple time series models, which would be particularly prone to underperform when there are sudden steep increases in cumulative death counts, and so the steep increases that we highlighted might partially explain our results. Furthermore, these cited studies were carried out during the early weeks of our study, and we would expect the compartmental models to have increased in sophistication over time and the model parameters and assumptions to have improved.

A major strength of our study is our source of data, which presented an opportunity to study the 'wisdom of the crowd', and provided the necessary conditions for the crowd being 'wise' [20] and without distortion, such as by social pressure [43] or restrictions against forecasting teams applying their own judgement [26]. These conditions include independent contributors, diversity of opinions, and a trustworthy central convener to collate the information provided [20]. Further strengths relating to the reliability of our findings arise from the high number of individual models. Our reported findings are limited to U.S. data and a particular set of models, and it is possible that different results may arise from other models, or for data from other locations, or other types of data, such as number of people infected. These are potential avenues for future research. Our ability to detect statistical differences was limited by the small sample sizes, with only 17 locations in each category, missing data, and a relatively short out-of-sample period.

It is suggested that relying on modelling alone leads to "missteps and blind spots", and that the best approach to support public policy decision making would involve a triangulation of insights from modelling with other information, such as analyses of previous outbreaks and discussions with frontline staff [44]. It is essential that modelling offers the most accurate forecasts. Probabilistic forecasts reflect the inherent uncertainties in prediction. Although individual models can sometimes be more accurate than combined methods, relying on forecasts from combined methods provides a more risk-averse strategy, as the best individual model will not be clear until records of historical accuracy are available, and the best performing model will typically change over time. At the start of an epidemic, when it is not clear which model has the best performance, the statistical expectation is that the average method will score far better than a model chosen randomly, or chosen on the basis of no prior history. This was the case at the start of the COVID-19 pandemic.

The existence of outlying forecasts presents challenges to forecast combining. These can arise due to model-based factors or factors involving the actual number of deaths. The former include computational model errors, which can happen occasionally, and model assumptions being incorrect, which will typically apply in the early stages of a pandemic. The latter factors include data updates and changes in definitions. Some models can be adapted to allow for data anomalies. The removal of outlying forecasts may be added to the pre-combining screening process, but screening criteria for outliers may be arbitrary and it will be subjective. A more objective way to tackle outlying forecasts is to use the median combination, and that was the approach taken by the COVID-19 Hub in July 2021, having previously relied on a mean ensemble. Our earlier study suggested that factoring historical accuracy into forecast combinations may achieve greater accuracy than the median combination [32]. Both our studies have involved the use of performance-weighted mean methods, and our current study has shown

that they are not sufficiently robust to outliers. We recommend further research into weighted methods and the effect of model type on the relative performance of combined methods.

## Supporting information

**S1 Fig. Data availability for forecasts of cumulative COVID-19 deaths.** [*] Based on information recorded on the COVID19 Hub with citations as recorded on 25/2/22; [a] Only provided forecasts of numbers of cumulative COVID-19 deaths; [b] Only provided forecasts of numbers of incident COVID-19 deaths.
(TIF)

**S1 Table. Individual forecasting models.**
(PDF)

**S2 Table. For incident mortality, 95% interval MIS and MWIS for each prediction horizon.** Lower values are better. [a] best method for each horizon in each column; [b] score is significantly lower than the mean combination; [c] score is significantly lower than the median combination.
(PDF)

**S3 Table. For incident mortality, calibration for all locations.**
(PDF)

**S4 Table. For incident mortality, calibration for U.S.**
(PDF)

**S5 Table. For incident mortality, calibration for high mortality locations.**
(PDF)

**S6 Table. For incident mortality, calibration for medium mortality locations.**
(PDF)

**S7 Table. For incident mortality, calibration for low mortality locations.**
(PDF)

**S8 Table. For cumulative mortality, 95% interval MIS and MWIS for each prediction horizon.** Lower values are better. [a] best method for each horizon in each column; [b] score is significantly lower than the mean combination; [c] score is significantly lower than the median combination.
(PDF)

**S9 Table. For cumulative mortality, calibration for all locations.**
(PDF)

**S10 Table. For cumulative mortality, calibration for U.S.**
(PDF)

**S11 Table. For cumulative mortality, calibration for high mortality locations.**
(PDF)

**S12 Table. For cumulative mortality, calibration for medium mortality locations.**
(PDF)

**S13 Table. For cumulative mortality, calibration for low mortality locations.**
(PDF)

**S14 Table. Sensitivity analysis for incident mortality, skill scores of the 95% interval MIS and MWIS after excluding locations for which there were noticeable changes in reporting**

**patterns. S**hows percentages. Higher values are better. [a] best method in each column.
(PDF)

**S15 Table. Sensitivity analysis for cumulative mortality, skill scores of the 95% interval MIS and MWIS after excluding locations for which there were noticeable changes in reporting patterns.** Shows percentages. Higher values are better. [a] best method in each column.
(PDF)

# Acknowledgments

We thank Nia Roberts for helping us understand the license terms for the forecast data from the COVID-19 Forecast Hub. We also thank the anonymous reviewers for their comments.

# Author Contributions

**Conceptualization:** Kathryn S. Taylor, James W. Taylor.

**Data curation:** Kathryn S. Taylor.

**Formal analysis:** James W. Taylor.

**Funding acquisition:** Kathryn S. Taylor.

**Investigation:** Kathryn S. Taylor, James W. Taylor.

**Methodology:** James W. Taylor.

**Project administration:** Kathryn S. Taylor.

**Software:** Kathryn S. Taylor, James W. Taylor.

**Validation:** James W. Taylor.

**Visualization:** Kathryn S. Taylor.

**Writing – original draft:** Kathryn S. Taylor.

**Writing – review & editing:** Kathryn S. Taylor, James W. Taylor.

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
