## [Decision Letter · Decision Letter 0]

17 Jan 2022

PONE-D-21-30265Forecasts of weekly incident and cumulative COVID-19 mortality in the United States: A comparison of combining methodsPLOS ONE

Dear Dr. Taylor,

Thank you for submitting your manuscript to PLOS ONE. After careful consideration, we feel that it has merit but does not fully meet PLOS ONE’s publication criteria as it currently stands. Therefore, we invite you to submit a revised version of the manuscript that addresses the points raised during the review process.

All three reviewers have recognized the importance and timeliness of the topic. However, they have also highlighted several criticalities. Please refer to their detailed reviews for indications. Please be sure to answer all their comments in your revision.==============================

We look forward to receiving your revised manuscript.

Kind regards,

Maurizio Naldi

Academic Editor

PLOS ONE

Journal Requirements:

Reviewers' comments:

Reviewer's Responses to Questions

**Comments to the Author**

1. Is the manuscript technically sound, and do the data support the conclusions?

Reviewer #1: Partly

Reviewer #2: Yes

Reviewer #3: Yes

2. Has the statistical analysis been performed appropriately and rigorously? 

Reviewer #1: Yes

Reviewer #2: I Don't Know

Reviewer #3: Yes

3. Have the authors made all data underlying the findings in their manuscript fully available?

Reviewer #1: No

Reviewer #2: No

Reviewer #3: Yes

4. Is the manuscript presented in an intelligible fashion and written in standard English?

Reviewer #1: Yes

Reviewer #2: Yes

Reviewer #3: Yes

5. Review Comments to the Author

Reviewer #1: The manuscript addresses an interesting and timely topic. The use of ensamble approaches is sound and comparisons across methods is very useful in practice. Some comments follow.

1. I would emphasize the importance of quantifying the uncertainty associated with forecasts. Therefore I kindly ask to report more predictive quantiles (1%, 2.5%, 5%, 10%, . . . , 90%, 95%, 97.5%, 99%) in addition to their point forecasts. This motivates considering both forecasts of cumulative and incident quantities, as predictive quantiles for these generally cannot be translated from one scale to the other.

2. The use of the interval score (Gneting and Raftery, 2007) is sound. The three summands can be interpreted as a measure of sharpness and penalties for under- and overprediction, respectively. I strongly believe that the weighted interval score (Bracher, J., Ray, E. L., Gneiting, T., and Reich, N. G. (2020a). Evaluating epidemic forecasts in an interval

format. PLOS Computational Biology) shoul be considered. It combines the absolute error of the

predictive median and the interval scores achieved for the nominal levels. It is a well-known

quantile-based approximation of the continuous ranked probability score.

3. Maybe I miss something, or there is something swept under the carpet. In any ensamble-based approach, the choice of the weights is crucial. Please, provide more details on this point.

4. It would also be interesting to see a discussion of the performance of the models belonging to different families. Ioannidis et al (Ioannidis JPA, Cripps S, Tanner MA. Forecasting for COVID-19 has failed. Int J Forecast. 2020. http://www.sciencedirect.com/science/article/pii/S016920%7020301199) discussed that some approaches, mainly those referring to the SIR family, have failed in providing reasoable forecasts. On the other hand, data-driven approaches show much better performances in forecasting the evolution of the epidemic (see e.g. Mingione, M., Di Loro, P. A., Farcomeni, A., Divino, F., Lovison, G., Maruotti, A., & Lasinio, G. J. (2021). Spatio-temporal modelling of COVID-19 incident cases using Richards’ curve: An application to the Italian regions. Spatial Statistics, 100544; Girardi, P., Greco, L., & Ventura, L. (2021). Misspecified modeling of subsequent waves during COVID‐19 outbreak: a change‐point growth model. Biometrical Journal).

Reviewer #2: Review of “Forecasts of weekly incident and cumulative COVID-19 mortality in the United States: A comparison of combining methods”

Thank you for the opportunity to review this interesting paper. The topic is important. This analysis builds on a previously published study of combining methods for COVID-19 forecasting models in the United States and extends it using more data. The methods are interesting and seem mostly technically sound to me, although I think a comparison with percent errors is essential. I have a number of clarifying questions about the text. My main comment is about data and code availability. Pointing to the input data sources is not sufficient, in my opinion, as the authors must transform and manipulate the data to conduct the analysis. The devil is in the details with forecasting and predictive validity. The authors should publish their code and data in a public repository to facilitate a complete review of the research. That is the standard in the field for this kind of work (all other papers I’ve read on the topic have done so) and also required by the journal for publication: https://journals.plos.org/plosone/s/materials-software-and-code-sharing.

Comments:

Abstract

1. Methods section is vague. How specifically was the analysis conducted? “comparing accuracy” is not specific. Which metrics were used? How were they calculated? How did you design your holdouts? The basic details should be clear in the abstract.

2. The results section is also vague. Specific values should be cited. By what margin much did the best performing models overtake the others?

3. Conclusions: please indicate how this study did or did not concur with the previous study. What more can we learn from adding additional data?

Methods

4. Why was only 1 year of data used when there are nearly 2 years of data available now since the start of the pandemic?

5. I am not sure that evaluating absolute error makes the most sense, as it biases the results towards higher mortality moments and locations. I would like to see a comparison to relative error metrics, such as the median absolute percent error. This has also been done in prior analyses so would be standard.

6. I would recommend placing the trimming and other details of how forecasts are combined in the main text, given journal length limits, and how central those details are to the work at hand.

Results

7. How did the methods you describe here perform in comparison to the forecasts hub’s own internal ensemble model? Please highlight this important comparison.

Reviewer #3: Thank you for the opportunity to review Taylor and Taylor’s manuscript that compares different methods of combining COVID-19 mortality forecasts. Their research is an important contribution to the forecasting field. Analysis of ensembling approaches is sorely needed; and it is also timely as the COVID-19 epidemic continues to change and forecasts continue to be used for operational decision making. Moreover, but as the field continues to move towards open and collaborative science, their work has direct application to future forecasting. Their clearly written manuscript has many strengths – it compares simple and more complex methods; it harnesses a year’s worth of data for evaluation; and focuses analyses on probabilistic distributions. In addition, there are also opportunities to improve to the text and analysis (see comments below) .

MAJOR COMMENTS

Abstract

Line 30: I suggest cutting the text about “extended and new datasets”. It’s not apparently clear to readers what the authors are referring to if they haven’t read their previous manuscript.

Line 35: The COVID-19 Forecast Hub collects probability distributions as well as point forecasts. The authors are referring here to the 50% quantile of probability distributions as ‘point forecasts’. Please refer to them throughout the manuscript as the 50% quantile, rather than a point forecast. [Also, see comments under Material and Methods, as I recommend excluding all 50% quantile analyses. These forecasts are 1) not useful for outbreak response, and 2) misleading in terms of communicating uncertainty.]

Lines 35-37: Please list the evaluation metrics (mean interval score) and the combination methods here.

Line 38 and 39: The first sentence of the Results is very confusing. Does the ‘average performance of these models’ refer to the Mean method, the Ensemble model, or something else?

Line 46: How do you define “sufficient”? Length of historical data was beyond the scope of the analysis, but the text implies this was considered within the analysis.

Introduction

Line 89: Recommend not referring to reported data as ‘delayed’. ‘Reporting patterns’ is more accurate to describe the nature of the changes to the CSSE datasets and more accurately describes the descriptive nature of this part of the analysis.

Materials and Methods

Lines 109-110: The number of the week’s is very confusing here. What is the first week of the epidemic, and are you referring to the first week in the US? Because forecasts were not collected at ‘week 1’ of outbreak, I suggest starting the start date and the end date, with the start date referred to as ‘week 1 of the analysis’.

Lines 113-116: I strongly recommend changing the inclusion criteria for this analysis. Models that were not included in the COVID-19 Forecast Hub should also be excluded here. This will help with comparability between the ensemble approaches. The COVID-19 Forecast Hub excludes forecasts that are improbable, such as if the number of cases or deaths exceeds the population size, not based on their predictions being ‘too large’. Details are provided here: https://covid19forecasthub.org/doc/ensemble/

Line 122: In the evaluation, did you use the CSSE reported counts at the end of the analysis period? OR was each date evaluated against the data available at the time. Please clarify in the methods.

Line 122: What is the rationale for focusing only on the 95% PIs? Information can be gained be examining all intervals (7 available) and weighting them (a method applied by Cramer et al, 2021: https://www.medrxiv.org/content/10.1101/2021.02.03.21250974v3)

Line 122: Because the interval score is a combination of calibration and sharpness, I wonder if the authors considered presenting all three metrics – calibration, sharpness, and IS? This might provide additional insights and if the authors are only presenting metrics at alpha of 0.05, they have the space to do so.

Line 130: The field is moving away from communication of single numbers and towards ranges. Point predications are rarely used to communicate forecasts in the COVID-19 pandemic, and generally discouraged. Thus, I don’t find the point forecast analysis to be useful and suggest removing it.

Line 142-146: Changes in the reported death counts were not always due to reporting delays. For example, the large spike observed in the winter 2020/2021 in Ohio reflects a change in how the state defined a death. Even the everchanging landscape of the pandemic, I’d recommend referring to these anomalies as “reporting patterns”, and perhaps defining examples of backlogged deaths, reporting dumps, or changes in definitions.

Lines 157-159: Please define ‘overconfidence’ and ‘underconfidence’ and describe how they relate to the various trimming methods.

Line 161: Can you say more about the ‘previous best’? It’s not clear to me why you added this model or if reference 42 is the correct reference for is. What does this add to the analysis?

Line 171-172: Inclusion of individual models aids in the overall point that combining is better, however, the inclusion criteria is pretty strict here. Several models were consistent submitters to the COVID-19 Forecast Hub but missed a week here or there. Consider broadening this inclusion criteria.

Line 171-172: I’d like to confirm that COVIDhub Baseline model was not included in this set? It’s not designed to be a true forecast but is rather a comparator point for the submitted models.

Results

Line 190: What were the thresholds used for the categories? How many states were in each category? Please also add to the discussion the limitation of not including a time component to the analysis, as the US experienced spatial heterogeneity in the outbreak and even lower incidence states had peaks, when model performance was subpar.

Line 230: Please note which statistical test you are referring to here.

Line 248: I think OK and WV are missing the dashed lines? If not, then there are no differences in reporting patterns overtime

Line 255: Please present these data in either points or bard. Lines implies that the data are longitudinal.

Line 266: Caution with describing Ohio and the individual models here. Because I don’t know which team model 33 is, I can’t speak to the accuracy of the text. It should be noted that several teams noted the spike in reported deaths in Ohio and assumed it to be an error, while other teams assumed it to be truth. Because this nuance is not available here, I recommend deleting mention of the individual models from the text.

Lines 278-281: Can you share that sensitivity analysis as a supplement?

Discussion

Line 298: The main limitation of the analysis is the lack of temporal analysis. The epidemic varied over time and space in the US, and consequently, so did the forecast performance. While I do not think that the authors need to include temporal analysis, they should include this as a limitation in the Discussion.

Line 302: As written, this implies that forecast type and timing was assessed in the analysis. Please provide a citation since this was beyond the scope of the manuscript.

Line 337: Same comments as line 302. Please reference.

MINOR COMMENTS

Line 61: Source 15 has been published; please update the reference.

Line 73: Center, in CDC, is spelled incorrectly.

Line 93 and 98: Reference 33 is incorrect. It should be reference 26 here, as reference 33 refers to the reported data, not the forecast data or the collaborations surrounding the forecast data.

Line 130 and 131: The sentence about point forecasts should be a new paragraph

6. PLOS authors have the option to publish the peer review history of their article (what does this mean?). If published, this will include your full peer review and any attached files.

Reviewer #1: No

Reviewer #2: No

Reviewer #3: No

---

## [Author Response · Author response to Decision Letter 0]

27 Feb 2022

Reply to comments of the Editor

The Editor writes: “Thank you for submitting your manuscript to PLOS ONE. After careful consideration, we feel that it has merit but does not fully meet PLOS ONE’s publication criteria as it currently stands. Therefore, we invite you to submit a revised version of the manuscript that addresses the points raised during the review process.

All three reviewers have recognized the importance and timeliness of the topic. However, they have also highlighted several criticalities. Please refer to their detailed reviews for indications. Please be sure to answer all their comments in your revision.”

Our response:

We are very grateful to the Editor for giving us the opportunity to revise our manuscript. We are also very appreciative of the reviewers for devoting time to read and comment on our manuscript. Extensive changes have been made to address the issues raised by the reviewers, involving the data, study design and reporting of results. We feel that the review process has very much improved our manuscript. 

We would like to point out to the reviewers that we now report the results for incident deaths and cumulative deaths separately. Please also note that we have changed the title so that we now refer to “interval forecasts” instead of simply “forecasts”, as we feel this more clearly describes the focus of our revised manuscript.

Reply to Reviewer #1

The reviewer writes: “The manuscript addresses an interesting and timely topic. The use of ensamble approaches is sound and comparisons across methods is very useful in practice. Some comments follow.”

Our response: We thank the reviewer for these positive comments.

The reviewer writes: “I would emphasize the importance of quantifying the uncertainty associated with forecasts. Therefore I kindly ask to report more predictive quantiles (1%, 2.5%, 5%, 10%, . . . , 90%, 95%, 97.5%, 99%) in addition to their point forecasts. This motivates considering both forecasts of cumulative and incident quantities, as predictive quantiles for these generally cannot be translated from one scale to the other.”

Our response: We now report more predictive quantiles by considering all the (symmetric) interval forecasts that can be constructed from the 23 quantiles. In response to other reviewers’ comments, we have removed the analysis of point (50% quantile) forecasts. Given our focus now on interval forecasts, we have added “Interval” to the title, so it is it is now 

“Interval forecasts of weekly incident and cumulative COVID-19 mortality in the United States: A comparison of combining methods”

The reviewer writes: “The use of the interval score (Gneting and Raftery, 2007) is sound. The three summands can be interpreted as a measure of sharpness and penalties for under- and overprediction, respectively. I strongly believe that the weighted interval score (Bracher, J., Ray, E. L., Gneiting, T., and Reich, N. G. (2020a). Evaluating epidemic forecasts in an interval format. PLOS Computational Biology) shoul be considered. It combines the absolute error of the predictive median and the interval scores achieved for the nominal levels. It is a well-known quantile-based approximation of the continuous ranked probability score.”

Our response: In addition to the 95% interval score, we now also report the weighted interval score, as we are now evaluating all (symmetric) intervals constructed from the 23 quantiles. As we take the average of the weighted interval score across the out-of-sample period, and across forecast horizons, in the paper, we refer to the “mean weighted interval score (MWIS)”.

The reviewer writes: “Maybe I miss something, or there is something swept under the carpet. In any ensamble-based approach, the choice of the weights is crucial. Please, provide more details on this point.”

Our response: The descriptions of the two weighted methods and all other combining methods were in a supplementary file in our previous submission. We appreciate that this important information should have been included in the main text, and so we have now done this. The combining methods are described in the section entitled “Forecast combining methods”.

The reviewer writes: “It would also be interesting to see a discussion of the performance of the models belonging to different families. Ioannidis et al (Ioannidis JPA, Cripps S, Tanner MA. Forecasting for COVID-19 has failed. Int J Forecast. 2020.

http://www.sciencedirect.com/science/article/pii/S016920%7020301199) discussed that some approaches, mainly those referring to the SIR family, have failed in providing reasoable forecasts. 

On the other hand, data-driven approaches show much better performances in forecasting the evolution of the epidemic (see e.g. Mingione, M., Di Loro, P. A., Farcomeni, A., Divino, F., Lovison, G., Maruotti, A., & Lasinio, G. J. (2021). Spatio-temporal modelling of COVID-19 incident cases using Richards’ curve: An application to the Italian regions. Spatial Statistics, 100544; Girardi, P., Greco, L., & Ventura, L. (2021). Misspecified modeling of subsequent waves during COVID‐19 outbreak: a change‐point growth model. Biometrical Journal).”

Our response: 

We now also consider the forecasting accuracy for compartmental/SEIR models versus non-compartmental models. We report the results in the sections entitled “Performance by model type for incident deaths” and “Performance by model type for cumulative deaths”.

We thank the reviewer for highlighting these references, which we now cite in the paper. In particular, the text in the discussion section now reads (4th para): 

“Previous studies have found that data driven models can perform better than compartmental models in forecasting COVID-19 data [9, 41, 42]. For forecasts of both mortalities, in many cases we found that there was no benefit in including non-compartmental models in a combination with compartmental models for a number of combining methods, including the mean combination. Non-compartmental models include simple time series models, which would be particularly prone to underperform when there are sudden steep increases in cumulative death counts, and so the steep increases that we highlighted might partially explain our results. Furthermore, these cited studies were carried out during the early weeks of our study, and we would expect the compartmental models to have increased in sophistication over time, and the model parameters and assumptions to have improved.”

Reply to Reviewer #2

The reviewer writes: “Thank you for the opportunity to review this interesting paper. The topic is important. This analysis builds on a previously published study of combining methods for COVID-19 forecasting models in the United States and extends it using more data. The methods are interesting and seem mostly technically sound to me, although I think a comparison with percent errors is essential.”

Our response: 

We thank the reviewer for their positive comments. 

We now report percentage errors in the form of skill scores. In the section entitled “Evaluating the interval forecasts” (1st para), we define the mean interval score (MIS) and mean weighted interval score (MWIS), and then later in this section (3rd para), we define skill scores. We write:

“We present results of the forecast accuracy evaluation in terms of the 95% interval MIS, MWIS, ranks and skill scores, which are calculated as the percentage by which a given method is superior to the mean combination, which is a common choice of benchmark in combining studies.”

The reviewer writes: “I have a number of clarifying questions about the text. My main comment is about data and code availability. Pointing to the input data sources is not sufficient, in my opinion, as the authors must transform and manipulate the data to conduct the analysis. The devil is in the details with forecasting and predictive validity. The authors should publish their code and data in a public repository to facilitate a complete review of the research. That is the standard in the field for this kind of work (all other papers I’ve read on the topic have done so) and also required by the journal for publication: https://journals.plos.org/plosone/s/materials-software-and-code-sharing.”

Our response: We now provide access to all our code files at https://doi.org/10.5281/zenodo.6300524. The README document explains the process of importing the publically available data from the COVID-19 Forecast Hub and the data preparation processes, which were carried out in Stata, before importing the data into GAUSS all generate the results of combining. These code files can be read using a text editor such as Notepad. The data is not ours to give. Our data availability statement now reads:

“Data were downloaded from the public GitHub data repository of the COVID-19 Hub at

https://github.com/reichlab/covid19-forecast-hub. The code used to generate the results is publically available on Zenodo at https://doi.org/10.5281/zenodo.6300524.”

Abstract

The reviewer writes: “Methods section is vague. How specifically was the analysis conducted? “comparing accuracy” is not specific. Which metrics were used? How were they calculated? How did you design your holdouts? The basic details should be clear in the abstract.”

Our response: The methods section of our abstract has been extended. It now provides more detail about the forecasting methods and their evaluation. We also now state the length of the out-of-sample periods. The text now reads: 

“We considered weekly interval forecasts, for 1- to 4-week prediction horizons, with out-of-sample periods of approximately 18 months ending on 8 January 2022, for multiple locations in the United States, using data from the COVID-19 Forecast Hub. Our comparison involved simple and more complex combining methods, including methods that involve trimming outliers or performance-based weights. Prediction accuracy was evaluated using interval scores, weighted interval scores, skill scores, ranks, and reliability diagrams.” 

The reviewer writes: “The results section is also vague. Specific values should be cited. By what margin much did the best performing models overtake the others?”

Our response: We thank the reviewer for highlighting this issue. The abstract now reads:

“The weighted inverse score and median combining methods performed best for forecasts of incident deaths. Overall, the leading inverse score method was 12% better than the mean benchmark method in forecasting the 95% interval and, considering all interval forecasts, the median was 7% better than the mean. Overall, the median was the most accurate method for forecasts of cumulative deaths. Compared to the mean, the median’s accuracy was 65% better in forecasting the 95% interval, and 43% better considering all interval forecasts.”

The reviewer writes: “Conclusions: please indicate how this study did or did not concur with the previous study. What more can we learn from adding additional data?”

Our response: We have had to be concise in our conclusions, given the word limit for the abstract. We now state: 

“The relative performance of combining methods depends on the extent of outliers”. 

This is explained more fully in the discussion section of the main text where the text reads (2nd para): 

“In our earlier study, the inverse score methods were the most accurate overall and the mean generally outperformed the median. The mean was also competitive against the inverse score method for many locations. The results of the current study for forecasts of cumulative deaths were not consistent with those of the earlier study, although much better results were achieved for the mean by combining only compartmental models, and when combining forecasts of incident deaths from all models, the relative performance of the inverse mean methods was considerably better. In the current study of cumulative deaths, the leading methods were generally the median methods and symmetric mean (for which the median is an extreme case). These methods are robust to outliers. The results of our two studies illustrate that, particularly for forecasts of cumulative deaths, the relative performance of combining methods depends on the extent of outlying forecasts, and that outlying forecasts were clearly more prevalent in the dataset for the current study”

Methods

The reviewer writes: “Why was only 1 year of data used when there are nearly 2 years of data available now since the start of the pandemic?”

Our response: We have updated the two datasets. In response to another reviewer we are now only using forecasts that passed the Hub’s screening tests and we used all available forecasts projected from forecast origins up to 8 January 2022.

The reviewer writes: “I am not sure that evaluating absolute error makes the most sense, as it biases the results towards higher mortality moments and locations. I would like to see a comparison to relative error metrics, such as the median absolute percent error. This has also been done in prior analyses so would be standard.”

Our response: In response to comments by the other reviewers, we have replaced the analysis of point (50% quantile) forecasts with the analysis of all the prediction intervals, evaluated using the weighted interval score. We now report percentage errors in the form of skill scores. In the section entitled “Evaluating the interval forecasts” (1st para), we define the mean interval score (MIS) and mean weighted interval score (MWIS), and then later in this section (3rd para), we define skill scores. We write:

“We present results of the forecast accuracy evaluation in terms of the 95% interval MIS, MWIS, ranks and skill scores, which are calculated as the percentage by which a given method is superior to the mean combination. The mean is a common choice of benchmark in combining studies.”

The reviewer writes: “I would recommend placing the trimming and other details of how forecasts are combined in the main text, given journal length limits, and how central those details are to the work at hand.”

Our response: We have followed the reviewer’s recommendation. The supplementary file on forecasting methods has been removed, and we describe each combining method in the section entitled “Forecast combining methods”.

Results

The reviewer writes: “How did the methods you describe here perform in comparison to the forecasts hub’s own internal ensemble model? Please highlight this important comparison.”

Our response: The analysis is now more focused on the Hub’s ensemble as, following the advice from another reviewer, our study now only includes forecasts that are included in the Hub’s ensemble. We were unable to reproduce perfectly the performance of the Hub’s ensemble (its performance should have been identical to that of the median in the post-sample period), which suggests that some forecasts were removed since they were submitted to and assessed by the Hub. We have reported the results based on the data that was available and we have referred to the Hub ensemble and the median collectively as the “median methods”. 

Reply to Reviewer #3

The reviewer writes: “Thank you for the opportunity to review Taylor and Taylor’s manuscript that compares different methods of combining COVID-19 mortality forecasts. Their research is an important contribution to the forecasting field. Analysis of ensembling approaches is sorely needed; and it is also timely as the COVID-19 epidemic continues to change and forecasts continue to be used for operational decision making. Moreover, but as the field continues to move towards open and collaborative science, their work has direct application to future forecasting. Their clearly written manuscript has many strengths – it compares simple and more complex methods; it harnesses a year’s worth of data for evaluation; and focuses analyses on probabilistic distributions. In addition, there are also opportunities to improve to the text and analysis (see comments below).”

Our response: We thank the reviewer for these positive comments.

MAJOR COMMENTS

Abstract

The reviewer writes: “Line 30: I suggest cutting the text about “extended and new datasets”. It’s not apparently clear to readers what the authors are referring to if they haven’t read their previous manuscript”

Our response: We have removed this sentence from the abstract, as recommended. 

The reviewer writes: “Line 35: The COVID-19 Forecast Hub collects probability distributions as well as point forecasts. The authors are referring here to the 50% quantile of probability distributions as ‘point forecasts’. Please refer to them throughout the manuscript as the 50% quantile, rather than a point forecast. [Also, see comments under Material and Methods, as I recommend excluding all 50% quantile analyses. These forecasts are 1) not useful for outbreak response, and 2) misleading in terms of communicating uncertainty.]”

Our response: Following the reviewer’s advice, we have now removed from the analysis the evaluation of 50% quantile forecasts. 

The reviewer writes: “Lines 35-37: Please list the evaluation metrics (mean interval score) and the combination methods here.”

Our response: The methods section of our abstract now provides more detail about the forecasting methods and their evaluation. The text now reads: 

“Our comparison involved simple and more complex combining methods, including methods that involve trimming outliers or performance-based weights. Prediction accuracy was evaluated using interval scores, weighted interval scores, skill scores, ranks, and reliability diagrams.” 

The reviewer writes: “Line 38 and 39: The first sentence of the Results is very confusing. Does the ‘average performance of these models’ refer to the Mean method, the Ensemble model, or something else?”

Our response: In our previous submission, this sentence referred to the average performance of all the models, which was calculated by considering the performance of each model individually and then averaging. We showed that this average performance did not exceed the performance of the mean combination (the ‘performance of the average was better than the average performance’). The dataset provided the opportunity to illustrate this standard result, which provides a motivation for combining. We have removed this analysis, and instead, summarise the results of the individual models included in our comparison in different ways in Tables 6 and 12. We hope that this is a more informative approach to illustrate that combining is better. 

The reviewer writes: “Line 46: How do you define “sufficient”? Length of historical data was beyond the scope of the analysis, but the text implies this was considered within the analysis.”

Our response: We appreciate the point made by the reviewer, and in revising the text we have removed that particular sentence. 

Introduction

The reviewer writes: “Line 89: Recommend not referring to reported data as ‘delayed’. ‘Reporting patterns’ is more accurate to describe the nature of the changes to the CSSE datasets and more accurately describes the descriptive nature of this part of the analysis.”

Our response: We have followed the reviewer’s recommendation by now referring to reporting “patterns” instead of “delays”. 

Materials and Methods

The reviewer writes: “Lines 109-110: The number of the week’s is very confusing here. What is the first week of the epidemic, and are you referring to the first week in the US? Because forecasts were not collected at ‘week 1’ of outbreak, I suggest starting the start date and the end date, with the start date referred to as ‘week 1 of the analysis’.”

Our response: We accept the reviewer’s point that referring to epidemic week was confusing. In our revised paper, we now avoid week numbering and simply refer to the actual dates of forecast origins. We do this in the text and in the labelling of figures. In the methods subsection “Dataset (2nd para), we state the start and end dates of our data: 

“Our dataset included forecasts projected from forecast origins at midnight on Saturdays between 9 May 2020 to 8 January 2022 for forecasts of cumulative COVID-19 deaths (88 weeks of data), and between 6 June 2020 and 8 January 2022 for forecasts of incident deaths (84 weeks of data).”

The reviewer writes: “Lines 113-116: I strongly recommend changing the inclusion criteria for this analysis. Models that were not included in the COVID-19 Forecast Hub should also be excluded here. This will help with comparability between the ensemble approaches. The COVID-19 Forecast Hub excludes forecasts that are improbable, such as if the number of cases or deaths exceeds the population size, not based on their predictions being ‘too large’. Details are provided here: https://covid19forecasthub.org/doc/ensemble/”

Our response: We thank the reviewer for sending the link to the Hub ensemble inclusion criteria. We accept that including only forecasts that were included by the Hub would provide a better comparison and greater consistency in our comparison. We have followed the reviewer’s recommendation. We were unable to reproduce perfectly the performance of the Hub ensemble (its performance should have been identical to that of the median in the post-sample period) which suggests that some forecasts were removed since they were submitted and assessed by the Hub. We have reported the results based on the data that was available and we have referred to the Hub ensemble and the median collectively as the ‘median methods’. The text in the methods subsection “Dataset” now reads (end of 1st para): 

“We only included forecasts that passed the Hub’s screening tests”.

The reviewer writes: “Line 122: In the evaluation, did you use the CSSE reported counts at the end of the analysis period? OR was each date evaluated against the data available at the time. Please clarify in the methods.”

Our response: We have clarified this point in the methods. The first sentence in the subsection of methods entitled “Evaluating the interval forecasts” the text now reads: 

“We evaluated out-of-sample prediction accuracy and calibration, with reference to the reported death counts on 15 January 2022, thus producing a retrospective evaluation.”

The reviewer writes: “Line 122: What is the rationale for focusing only on the 95% PIs? Information can be gained be examining all intervals (7 available) and weighting them (a method applied by Cramer et al, 2021: https://www.medrxiv.org/content/10.1101/2021.02.03.21250974v3)”

Our response: We now include in our analysis all (symmetric) intervals constructed from the 23 quantiles. 

The reviewer writes: “Line 122: Because the interval score is a combination of calibration and sharpness, I wonder if the authors considered presenting all three metrics – calibration, sharpness, and IS? This might provide additional insights and if the authors are only presenting metrics at alpha of 0.05, they have the space to do so.”

Our response: For the 95% interval forecasts, and for all intervals considered together, we are now presenting results for mean interval score (MIS), mean weighted interval score (MWIS), mean ranks (as the statistical tests are based on these), and skills scores (in response to comments by another reviewer). We chose to summarise the performance over time, performance by type of model and sensitivity analysis by using skill scores, as we believe they provide a useful way of measuring how much the performance of each method differs from the established benchmark mean method (which is often hard to beat), and how the methods compare with the leading methods. 

The reviewer writes: “Line 130: The field is moving away from communication of single numbers and towards ranges. Point predications are rarely used to communicate forecasts in the COVID-19 pandemic, and generally discouraged. Thus, I don’t find the point forecast analysis to be useful and suggest removing it.”

Our response: Following the reviewer’s recommendation, we have removed 50% quantile (point) forecasts from the paper. 

The reviewer writes: “Line 142-146: Changes in the reported death counts were not always due to reporting delays. For example, the large spike observed in the winter 2020/2021 in Ohio reflects a change in how the state defined a death. Even the everchanging landscape of the pandemic, I’d recommend referring to these anomalies as “reporting patterns”, and perhaps defining examples of backlogged deaths, reporting dumps, or changes in definitions.”

Our response: We acknowledge that changes in reporting patterns were not solely attributed to reporting delays and we have followed the reviewer’s recommendation by referring to “reporting patterns”. Please also see our response to this reviewer’s comments regarding line 248 and the states OK and WV.

The reviewer writes: “Lines 157-159: Please define ‘overconfidence’ and ‘underconfidence’ and describe how they relate to the various trimming methods.”

Our response: We decided to remove these terms, as we can see they are likely to cause confusion, and we thank the reviewer for highlighting this issue. Instead, we now refer to intervals that are overly narrow or overly wide. We do this when describing the exterior and interior trimming methods in the section entitled “Forecast combining methods” 

The reviewer writes: “Line 161: Can you say more about the ‘previous best’? It’s not clear to me why you added this model or if reference 42 is the correct reference for is. What does this add to the analysis?”

Our response: We include the ‘previous best’ forecast, as we feel it is natural to consider model selection as an alternative to model combination. The ‘previous best’ method simply selects the forecasts of the model that has the best historical accuracy (judged in terms of the interval score). The reference that we provide is the correct reference for this method. Capistrán and Timmermann refer to it on page 430 of their article Forecast combination with entry and exit. J Bus Stats 27(4): 428-440. We appreciate that the previous best is not a combining method, and so we have moved its introduction to the section entitled “Comparison with individual models”. To be consistent with this, we now position this method in the bottom row of the results tables, below the results of the combining methods. 

The reviewer writes: “Line 171-172: Inclusion of individual models aids in the overall point that combining is better, however, the inclusion criteria is pretty strict here. Several models were consistent submitters to the COVID-19 Forecast Hub but missed a week here or there. Consider broadening this inclusion criteria.”

Our response: We appreciate that our inclusion criteria may seem strict, but it would not be appropriate to include an individual model in our main results tables, along with the combining methods, unless forecasts were available from the individual model for all weeks in the out-of-sample period. The evaluation measures would not be comparable. However, we accept that it is interesting to provide, somehow, a comparison of accuracy of the individual models with the combining methods. To enable this, we now present summaries of the results of the individual models for which we had forecasts for at least half the out-of-sample period and at least half of the 52 locations. Tables 6 and 12 summarise skill scores based on scores calculated for the individual model and the benchmark method using only those weeks for which forecasts were available for the individual model. These tables and discussions of them are provided in the sections entitled “Performance of individual models for incident deaths” and “Performance of individual models for cumulative deaths”.

The reviewer writes: “Line 171-172: I’d like to confirm that COVIDhub Baseline model was not included in this set? It’s not designed to be a true forecast but is rather a comparator point for the submitted models.”

Our response: We concede that our reference to the inclusion of individual models in the comparison, in our previous submission, was confusing. In the analysis described in our revised manuscript we included forecasts that passed the Hub’s screening tests and that included forecasts from the COVID Hub baseline model. We believe that the reviewer was referring to the individual models that were included in our comparison. We can confirm that although the COVID Hub baseline model provided forecasts for more than half the locations and for more than half the out-of-sample period (the revised inclusion criteria for this analysis), we can confirm that we have not included this model in our comparison. In the section entitled “Comparison with individual models” the text reads:

“….we also summarise a comparison of the mean and median combinations with individual models for which we had forecasts for at least half the locations and at least half the out-of-sample period. Our inclusion criteria here is rather arbitrary, but the resulting analysis does help us compare the combining methods with the models of the more active individual teams. In this comparison, we excluded the COVID Hub baseline model, as it is only designed to be a comparator point for the models submitted to the Hub and not a true forecast”. 

Results

The reviewer writes: “Line 190: What were the thresholds used for the categories? How many states were in each category? 

Our response: We divided the 51 states into three equal groups of 17 states. We felt that setting thresholds would be arbitrary and splitting equally would be a pragmatic attempt to deal with the problem of scores for some locations dominating. We now report the number of states in each category. The text of the methods subsection “Evaluating the interval forecasts” now reads (3rd para): 

“We report results for the series of total U.S. deaths, as well as results averaged across all 52 locations. In addition, to avoid scores for some locations dominating, we also present results averaged for three categories, each including 17 states: high, medium and low mortality states. This categorisation was based on the number of cumulative COVID-19 deaths on 15 January 2022.”.

The reviewer writes: “Please also add to the discussion the limitation of not including a time component to the analysis, as the US experienced spatial heterogeneity in the outbreak and even lower incidence states had peaks, when model performance was subpar.”

Our response: To provide some insight into the potential change in ranking of methods over time, we now report the MWIS results separately for the first and second halves of the out-of-sample period. The text in “Evaluating the interval forecasts” (3rd para) reads:

“All results are for the out-of-sample period, and to provide some insight into the potential change in ranking of methods over time, we present MWIS results separately for the first and second halves of the out-of-sample period.”

These results are given in sections entitled “Changes over time in performance for incident deaths” and “Changes over time in performance for cumulative deaths”. 

The reviewer writes: “Line 230: Please note which statistical test you are referring to here.”

Our response: We now refer to the statistical test when presenting results in the sections entitled “Main results for incident deaths” (1st para) and “Main results for cumulative deaths (1st para)”. 

The reviewer writes: “Line 248: I think OK and WV are missing the dashed lines? If not, then there are no differences in reporting patterns overtime”

Our response: 

We can now see that our explanation was not clear. In the methods subsection “Evaluating the interval forecasts” (4th para) the text now reads:

 “We evaluated the effects of changes in reporting patterns on forecast accuracy. Changes in reporting patterns may involve reporting delays of death counts and changes in the definitions of COVID-19 deaths, both of which may lead to backdating of death counts and steep increases or decreases. Backdating of death counts would produce a problematic assessment in our retrospective evaluation of forecast accuracy, and sudden changes in death counts might cause some forecasting models to misestimate, particularly time series models. To obtain some insight, we compared reports of cumulative death counts for each location in files that were downloaded at multiple time points between 20 June 2020 and 15 January 2022. Locations for which there were notable effects of reporting patterns were excluded in sensitivity analysis.” 

In the results section entitled “Impact of changes in reporting patterns and outliers on forecast accuracy”(1st para) the text reads: 

“We observed changes in reporting patterns of death counts at 15 locations. Fig 6 shows examples of six locations where the changes were particularly noticeable. We found evidence of backdating in Delaware, Ohio, Rhode Island and Indiana. Backdating of historical death counts is shown as dashed lines. We noted a sharp drop in death counts in West Virginia in May 2021, suggesting a redefinition of COVID-19 deaths. There were sharp increases in death counts in Oklahoma in early April 2021 and in Delaware in late July 2021. We also observed sharp increases in death counts of two other locations, Missouri and Nebraska.” 

The reviewer writes: “Line 255: Please present these data in either points or bard. Lines implies that the data are longitudinal.”

Our response: Our reason for joining the lines was to aid visibility, but we accept that this was misleading. We have removed the lines and instead present just points in our revised figures (Figs 7 and 8). 

The reviewer writes: “Line 266: Caution with describing Ohio and the individual models here. Because I don’t know which team model 33 is, I can’t speak to the accuracy of the text. It should be noted that several teams noted the spike in reported deaths in Ohio and assumed it to be an error, while other teams assumed it to be truth. Because this nuance is not available here, I recommend deleting mention of the individual models from the text.”

Our response: We have redrafted this section and we no longer refer to individual models. 

The reviewer writes: “Lines 278-281: Can you share that sensitivity analysis as a supplement?”

Our response: The results of the sensitivity analyses are now presented as supplementary information (S14 Table and S15 Table).

Discussion

The reviewer writes: “Line 298: The main limitation of the analysis is the lack of temporal analysis. The epidemic varied over time and space in the US, and consequently, so did the forecast performance. While I do not think that the authors need to include temporal analysis, they should include this as a limitation in the Discussion.”

Our response: To provide some insight into the potential change in ranking of methods over time, we now report the MWIS results separately for the first and second halves of the out-of-sample period. The text in “Evaluating the interval forecasts” (3rd para) reads:

“All results are for the out-of-sample period, and to provide some insight into the potential change in ranking of methods over time, we present MWIS results separately for the first and second halves of the out-of-sample period.”

These results are given in sections entitled “Changes over time in performance for incident deaths” and “Changes over time in performance for cumulative deaths”. ”

The reviewer writes: “Line 302: As written, this implies that forecast type and timing was assessed in the analysis. Please provide a citation since this was beyond the scope of the manuscript.” and also “Line 337: Same comments as line 302. Please reference.”

Our response: We have removed the text as we can see it was unclear. 

MINOR COMMENTS

The reviewer writes: “Line 61: Source 15 has been published; please update the reference.”

Our response: We thank the reviewer for highlighting this point. The reference has been updated.

The reviewer writes: “Line 73: Center, in CDC, is spelled incorrectly.”

Our response: We thank the reviewer for highlighting this error. We have corrected the spelling. 

The reviewer writes: “Line 93 and 98: Reference 33 is incorrect. It should be reference 26 here, as reference 33 refers to the reported data, not the forecast data or the collaborations surrounding the forecast data.”

Our response: We thank the reviewer for highlighting this error. We have corrected the citations.

The reviewer writes: “Line 130 and 131: The sentence about point forecasts should be a new paragraph”

Our response: We now no longer discuss point forecasting.

---

## [Decision Letter · Decision Letter 1]

15 Mar 2022

Interval forecasts of weekly incident and cumulative COVID-19 mortality in the United States: A comparison of combining methods

PONE-D-21-30265R1

Dear Dr. Taylor,

We’re pleased to inform you that your manuscript has been judged scientifically suitable for publication and will be formally accepted for publication once it meets all outstanding technical requirements.

Kind regards,

Maurizio Naldi

Academic Editor

PLOS ONE

Additional Editor Comments (optional):

Reviewers' comments:

Reviewer's Responses to Questions

**Comments to the Author**

1. If the authors have adequately addressed your comments raised in a previous round of review and you feel that this manuscript is now acceptable for publication, you may indicate that here to bypass the “Comments to the Author” section, enter your conflict of interest statement in the “Confidential to Editor” section, and submit your "Accept" recommendation.

Reviewer #2: All comments have been addressed

Reviewer #3: All comments have been addressed

2. Is the manuscript technically sound, and do the data support the conclusions?

Reviewer #2: (No Response)

Reviewer #3: Yes

3. Has the statistical analysis been performed appropriately and rigorously? 

Reviewer #2: (No Response)

Reviewer #3: Yes

4. Have the authors made all data underlying the findings in their manuscript fully available?

Reviewer #2: (No Response)

Reviewer #3: Yes

5. Is the manuscript presented in an intelligible fashion and written in standard English?

Reviewer #2: (No Response)

Reviewer #3: Yes

6. Review Comments to the Author

Reviewer #2: (No Response)

Reviewer #3: Thank you to the authors for addressing all comments. In particular, the adjustments to include all quantiles, and thus better address uncertainty, have improved the manuscript. I have one very minor comment: The authors note that the COVID-19 Hub Ensemble method was switched to a median in July 2020. While this is correct, there was one additional change to the methods in November 20201. As of November 2021, the ensemble used a weighted approach based on WIS in 12 prior weeks (https://covid19forecasthub.org/doc/ensemble/). This is worth noting in the text.

7. PLOS authors have the option to publish the peer review history of their article (what does this mean?). If published, this will include your full peer review and any attached files.

Reviewer #2: No

Reviewer #3: No

---

## [Editor Report · Acceptance letter]

21 Mar 2022

PONE-D-21-30265R1 

Interval forecasts of weekly incident and cumulative COVID-19 mortality in the United States: A comparison of combining methods 

Dear Dr. Taylor:

I'm pleased to inform you that your manuscript has been deemed suitable for publication in PLOS ONE. Congratulations! Your manuscript is now with our production department. 

Kind regards, 

on behalf of

Professor Maurizio Naldi 

Academic Editor

PLOS ONE